# Constitutive activation of two-component systems reveals regulatory network interactions in *Streptococcus agalactiae*

Cosme Claverie [1,8], Francesco Coppolino [1,2,8], Maria-Vittoria Mazzuoli[1], Cécile Guyonnet[3,4,5], Elise Jacquemet [6], Rachel Legendre [6], Odile Sismeiro[1], Giuseppe Valerio De Gaetano[2], Giuseppe Teti[7], Patrick Trieu-Cuot[1], Asmaa Tazi[3,4,5], Concetta Beninati[2] & Arnaud Firon [1] ✉

Bacterial two-component systems (TCSs) are signaling modules that control physiology, adaptation, and host interactions. A typical TCS consists of a histidine kinase (HK) that activates a response regulator via phosphorylation in response to environmental signals. Here, we systematically test the effect of inactivating the conserved phosphatase activity of HKs to activate TCS signaling pathways. Transcriptome analyses of 14 HK mutants in *Streptococcus agalactiae*, the leading cause of neonatal meningitis, validate the conserved HK phosphatase mechanism and its role in the inhibition of TCS activity in vivo. Constitutive TCS activation, independent of environmental signals, enables high-resolution mapping of the regulons for several TCSs (e.g., SaeRS, BceRS, VncRS, DltRS, HK11030, HK02290) and reveals the functional diversity of TCS signaling pathways, ranging from highly specialized to interconnected global regulatory networks. Targeted analysis shows that the SaeRS-regulated PbsP adhesin acts as a signaling molecule to activate CovRS signaling, thereby linking the major regulators of host-pathogen interactions. Furthermore, constitutive BceRS activation reveals drug-independent activity, suggesting a role in cell envelope homeostasis beyond antimicrobial resistance. This study highlights the versatility of constitutive TCS activation, via phosphatase-deficient HKs, to uncover regulatory networks and biological processes.

Two-component systems (TCSs) are one of the main bacterial signalling mechanisms. In their simplest form, an environmental signal activates a histidine kinase (HK), which phosphorylates a cognate response regulator (RR), leading to the transcription of specific genes that mediate the cellular response to the stimuli[1]. Actually, TCSs are sophisticated molecular machinery with buffering and insulating mechanisms that

dynamically control specific or global cellular responses[2–4]. Considerable effort has been made to define TCS regulatory networks in both model and pathogenic species, including by comprehensive analysis[5,6]. Although knowledge gained in one species can provide information about homologous systems, TCS are characterised by their diversity, plasticity and evolvability[7,8]. This prevents global inferences even

[1]Institut Pasteur, Université Paris Cité, Department of Microbiology, Biology of Gram-Positive Pathogens, Paris, France. [2]University of Messina, Department of Human Pathology, Messina, Italy. [3]Assistance Publique-Hôpitaux de Paris, Hôpital Cochin, Department of Bacteriology, French National Reference Center for Streptococci, Paris, France. [4]Université Paris Cité, Institut Cochin, Institut National de la Santé et de la Recherche Médicale U1016, Centre National de la Recherche Scientifique UMR8104, Team Bacteria and Perinatality, Paris, France. [5]Fédération Hospitalo-Universitaire Fighting Prematurity, Paris, France. [6]Institut Pasteur, Université Paris Cité, Bioinformatics and Biostatistics Hub, Paris, France. [7]Scylla Biotech Srl, Messina, Italy. [8]These authors contributed equally: Cosme Claverie, Francesco Coppolino. ✉e-mail: arnaud.firon@pasteur.fr

between closely related species[9,10]. This evolution of regulatory networks is sustained by several mechanisms, including mutations, horizontal gene transfer, duplication followed by neofunctionalization, and rewiring that shapes adaptation and speciation[11–13].

Functional, evolutionary, and system analyses require characterising individual signalling pathways and integrating them into the cellular regulatory network. Traditionally, regulons are characterised using inactivated TCS mutants. One common pitfall is that TCSs are not active until the specific, but usually unknown, stimulus is provided. Current approaches to overcome signal requirements are based on phospho-mimetic mutation of the RR[14,15] and profiling of protein-DNA interaction[16–18]. An alternative approach exploits the distinct HK enzymatic activities. The HK cytoplasmic core called the transmitter module, is composed of the DHp (Dimerisation and Histidine phosphotransfer) and CA (catalytic and ATP-binding) domains[19–21]. The two domains are dynamically structured in specific conformations that catalyse three distinct reactions: autophosphorylation of a conserved histidine residue in the DHp domain, phosphotransfer to a conserved aspartate on the RR, and RR dephosphorylation. Pioneering studies have identified mutations abolishing the HK phosphatase activity leading to increased RR phosphorylation and signalling pathway activation[22–24].

The importance of HK phosphatase activity in vivo has been initially debated, especially when considering the lability of RR phosphorylation and spontaneous dephosphorylation rate[25]. Nowadays, phosphatase activity is recognised as essential for the dynamics of the response and to ensure that the RR is activated by the cognate HK only[26,27]. Co-evolving residues and HKs conformational rearrangements ensure the specificity and directionality of enzymatic reactions[28–30]. While the activation mechanism involving the conserved histidine residue is fundamentally conserved among HKs, the phosphatase mechanism has remained more elusive due to variations in the DHp domain[31]. Then, a seminal study proposed a conserved phosphatase mechanism for the two main HisKA and HisKA_3 families, identifying conserved motifs and specific catalytic residues needed for the correct positioning of nucleophilic attack[31,32]. Substitution of the catalytic residues abolishes the phosphatase activity without impacting the autokinase and phosphotransfer activities, resulting in increased RR phosphorylation and pathway activation for the individual HKs reported to date[32–37].

This study aims to systematically test the proposed conserved mechanism of phosphatase activity, the in vivo effect of phosphatase-deficient HK, and the activation of the regulatory network in all HisKA and HisKA_3 systems in a bacterium. We focused on *Streptococcus agalactiae* (Group B *Streptococcus*, GBS), a pathobiont that is commensal in adults but pathogenic during pregnancy and in neonates, for whom it is the leading cause of invasive infections[38,39]. We report that targeting HK phosphatase activity provides high-resolution views of signalling pathways for most TCSs independently from environmental signals. In addition, regulatory network activation resolves the connectivity between TCSs involved in host-pathogen interactions and reveals the physiological function of a TCS involved in antimicrobial resistance. This systematic analysis argues for the widespread adoption of this gain-of-function approach to decipher TCSs signalling in genetically manipulable species.

## Results

### The HK⁺ collection targets the phosphatase activity of Histidine Kinase

We undertook a genetic approach to systematically test the hypothesis of a conserved dephosphorylation mechanism in the two major HK families[32]. The genome of the BM110 strain belonging to the hypervirulent clonal complex 17 (CC-17) encodes 20 HKs[40], among which 12 and 2 have a HisKA and HisKA_3 DHp domain, respectively (Supplementary Data 1). Their H-box motif always contains the conserved phospho-acceptor histidine, immediately followed by the predicted

phosphatase motif (Fig. 1A). Eleven of the twelve HisKA proteins have the E/DxxT/N motif with a putative threonine catalytic residue, while the remaining HisKA protein (BceS) has a divergent sequence composition (QMKV) with a valine at the predicted catalytic position (Fig. 1A). The two HisKA_3 proteins have the DxxxQ/H motif with the predicted glutamine or histidine catalytic residue (Fig. 1A). The 14 HKs encoding genes are organised in operon with their cognate response regulator (RR) belonging to the OmpR (with HisKA) or LuxR (with HisKA_3) family, but one system is not functional (HK10655-RR10650$^{fs}$) due to a pseudogenization of the RR in the CC-17 hypervirulent GBS lineage (Supplementary Data 1).

We generated 14 strains, called the HK⁺ collection, with an alanine substitution of the predicted phosphatase catalytic residue (Fig. 1A). Whole-genome sequencing confirmed the chromosomal substitution of targeted base pairs and the absence of secondary mutations in 11 out of the 14 HK⁺ strains. In the three remaining HK⁺ (CovS$_{T282A}$, VicK$_{T221A}$, and RelS$_{T208A}$), we sequenced independent mutants and selected one with a single secondary mutation (Supplementary Data 2). Notably, the selected VicK$_{T221A}$ mutant possesses a non-synonymous polymorphism in the glutamine transporter GlnPQ, which we cannot dismiss as a potential compensatory mutation. Four independently constructed VicK$_{T221A}$ mutants exhibit putative compensatory mutations (Supplementary Data 2), a phenomenon commonly observed in mutants within the homologous WalRK system, which is essential for cell wall remodelling during growth and division[41–43].

Individual growth curves show a significant effect ($|F| > 0.1$, Mann Whitney test $p < 10^{-4}$) of the HK⁺ mutation for four mutants (Fig. 1B). The CovS$_{T282A}$ and CiaH$_{T228A}$ have a reproducible fitness advantage compared to the WT strain, while the VicK$_{T221A}$ and SaeS$_{T133A}$ have fitness defect. Notably, the slow-growing VicK$_{T221A}$ mutant is unstable and gives rise to faster-growing cultures, likely due to additional mutations, while the SaeS$_{T133A}$ mutant exhibits a density-dependent phenotype characterised by a decreasing growth rate in the exponential phase and a lower final OD (Supplementary Fig. 1). In addition, two mutants have increased antibiotic susceptibilities: the VicK$_{T221A}$ mutant against beta-lactams, in agreement with a conserved function in cell wall metabolism, and the RelS mutant against fosfomycin (Supplementary Data 3).

### HK⁺ activate positive feedback loops

To test TCS activation, we first relied on positive feedback loops. This autoregulation is often observed through direct transcriptional activation of the TCS operon by the activated RR[2]. We therefore analysed the transcription of all HKs and RRs encoding genes ($n = 41$, including non-HisKA and HisKA_3 TCSs and an orphan RR) in each HK⁺ mutant by RNA-sequencing from cultures grown in a standardised condition (THY: Todd Hewitt supplemented with 1% Yeast extract, pH 7.4, 37 °C, exponential growth phase OD$_{600}$ = 0.5). A positive feedback loop, indicated by a significant fold change greater than two relative to the WT strain ($\log_2$ FC > 1; p-adj < $10^{-4}$) for the HK and RR genes, is observed in seven HK⁺ mutants (Fig. 1C). Furthermore, two TCSs are significantly regulated in an unrelated HK⁺ mutant: the HK11050-RR11055 system, which does not contain a HisKA and HisKA_3 domain, in the VicK$_{T221A}$ mutant and the RelRS system, which is not positively auto-regulated, in the CiaH$_{T228A}$ mutant (Fig. 1C).

As an independent approach to test TCS activation, we introduced in each mutant a vector expressing an epitope-tagged copy of the cognate regulator. For two mutants (VicK$_{T221A}$ and RelS$_{T208A}$), an increased level of phosphorylation of the ectopically expressed regulator is detected in the HK⁺ mutant compared to the WT strain after Phos-Tag electrophoresis and western analysis with anti-FLAG antibodies (Fig. 1D). However, due to competition between WT and epitope-tagged regulators and variability in the stability of phosphorylated aspartate, no conclusions could be drawn for most regulators. This highlights the need to quantify the level of phosphorylation using

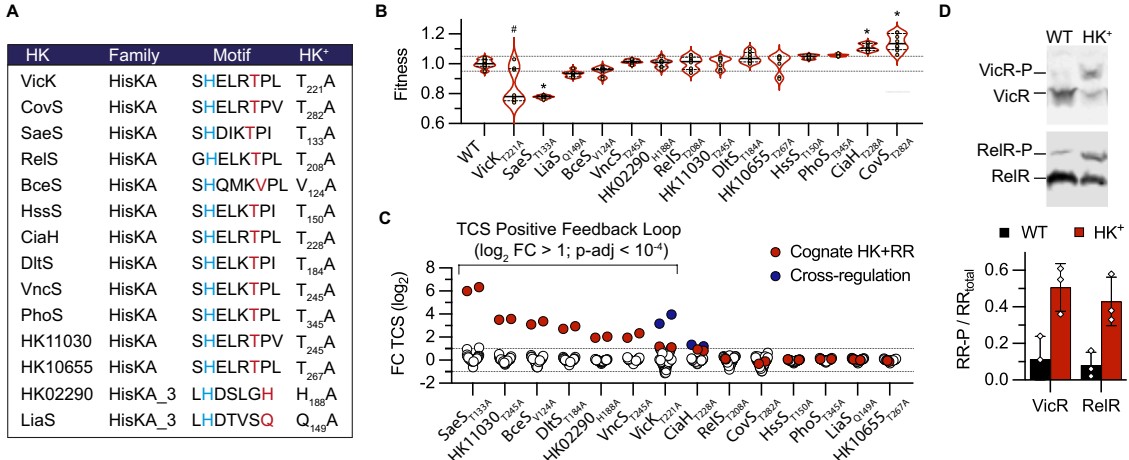

**Fig. 1 | Mutation of HK phosphatase catalytic residue activates TCS signalling.**
**A** Conserved motif of the HisKA and HisKA_3 histidine kinases with the phospho-acceptor histidine (blue) and the predicted residue specifically involved in the phosphatase activity (red). The phosphatase residue is substituted by an alanine in the HK+ mutants. Genes ID, proteins ID, and alternative names of TCSs are provided in Supplementary Data 1. **B** Fitness of HK+ mutants. The violin plots represent the distribution of the relative doubling time (Fitness (F) = doubling time WT mean / doubling time mutant) in an exponential growth phase in THY with the median (bar) and the interquartile range (dashed lines). Individual dots are shown for biological replicate ($n = 16$ for the WT, $n = 8$ for mutants), and significant differences are highlighted (*, $|F| > 0.1$, two-tailed Mann Whitney test $p < 10^{-4}$). The bimodal distribution due to the occurrence of faster-growing $VicK_{T221A}$ suppressors is highlighted (#). Corresponding growth curves and doubling times are shown in Supplementary Fig. 1. **C** Activation of transcriptional feedback loops in HK+

mutants. Fold changes (FC) for all genes encoding TCS ($n = 41$) in each HK+ mutant after RNA-seq analysis are shown as dots. The HK-RR gene pair in the corresponding HK+ mutant is highlighted in red (e.g., *saeRS* in $SaeS_{T133A}$). Cross-regulations, defined as significant differential expression of a TCS gene pair not corresponding to the HK+ mutation, are highlighted in blue (*hk11050-rr11055* in $VicK_{T221A}$ and *relRS* in $CiaH_{T228A}$). **D** Activation of the VicR and the RelR response regulators by phosphorylation in the corresponding HK+ mutants. Upper: representative Phos-Tag western-blots with anti-FLAG antibodies allowing to separate phosphorylated and non-phosphorylated forms of the ectopically expressed epitope-tagged RR. Bottom: quantification of the proportion of phosphorylated regulators in the WT (black) and the cognate HK+ mutant (red). Bars represent the mean with SD of biological replicate ($n = 3$). Source data are provided in Supplementary Data 4E for panel (**C**) and as a Source Data file for panels (**B**) and (**D**).

specific antibodies directed against each native RR. Overall, by considering epitope-tagged RR activation by phosphorylation and positive feedback loops, the majority (8/14) of HK+ mutations appear to activate the corresponding TCS signalling pathway.

## The activated gene regulatory network
To characterise the activated pathways, we analysed the RNA-seq profiles of each HK+ mutant grown under standardised conditions, independent of specific environmental cues (i.e., exponential phase in THY pH 7.4 at 37 °C). Statistical analysis of differentially expressed genes (DEGs: Supplementary Data 4) grouped the HK+ mutants into three main categories based on the adjusted $p$-values. Six HK+ mutants ($HK11030_{T245A}$, $VncS_{T245A}$, $SaeS_{T133A}$, $BceS_{V124A}$, $HK02290_{H188A}$, and $DltS_{T184A}$) show DEGs associated with striking statistical significance (p-adj $< 10^{-250}$), revealing the activated regulons with high resolution (Fig. 2A and Supplementary Fig. 2). Four additional mutants ($RelS_{T208A}$, $CiaH_{T228A}$, $VicK_{T221A}$, and $CovS_{T282A}$) show DEGs with lower statistical significance (p-adj $> 10^{-150}$), suggesting complex regulons or intermediate TCS activation (Supplementary Fig. 2). The remaining four mutants ($HssS_{T150A}$, $LiaS_{Q149A}$, $PhoS_{T345A}$, and the $HK10655_{T267A}$ with a frameshifted RR) gave no or low significant signals (p-adj $> 10^{-10}$) (Supplementary Fig. 2).

Since most RRs are transcriptional activators, we focused the analysis on activated genes. By applying strict thresholds (FC > 3, p-adj $< 10^{-4}$) for normalisation between samples and excluding genes with very low read counts in all samples and genes localised in mobile genetic elements, 219 genes (11.9% of the 1838 genes analysed) are transcriptionally activated in at least one HK+ mutant (Supplementary Data 4E). Transcriptional activation can be up to 8000-fold, with an average fold change of 61.6-fold and an uneven distribution between HK+ mutants (Fig. 2B).

The number of activated genes ranges from 3 ($HK11030_{T245A}$) to 139 ($VicK_{T221A}$) (Fig. 2B and Supplementary Data 4F). Five regulatory

systems activate a specific genetic programme, four of them ($HK11030_{T245A}$, $HK02290_{H188A}$, $VncS_{T245A}$, $DltS_{T184A}$) positively regulating a single functional genetic module composed of their own operon and at least one additional gene involved in the cellular response localised into, or adjacent to, the TCS operon (Fig. 2C), and one system ($CiaH_{T228A}$) coordinating the activation of at least six independent loci (Supplementary Fig. 3). Four additional TCSs activate specific genes but share 1 to 3 activated genes with the $VicK_{T221A}$ mutant (Fig. 2D). One of these connected systems ($HssS_{T150A}$) is specialised in haem detoxification via the transcriptional activation of the *hrtBA* genes encoding a specific ABC transporter[44], which is similarly activated in the $VicK_{T221A}$ mutant. The three additional connected systems activate several loci involved in host-pathogen interaction ($SaeS_{T133A}$: adhesins and secreted proteins), drug resistance ($BceS_{V124A}$: transporters and peptidase), or nucleotide metabolism ($RelS_{T208A}$: de novo purine synthesis and ectonucleotidases), with ($SaeS_{T133A}$, $BceS_{V124A}$) or without ($RelS_{T208A}$) a positive feedback loop (Fig. 2D and Supplementary Fig. 3).

## Positive and negative interaction between TCS systems
Overall, the HK+ mutation activates the signalling pathway for 10 out of 12 TCSs (Fig. 2D), excluding the $CovS_{T282A}$ repressing system analysed separately and the negative control $HK10655_{T267A}$ with a frameshifted RR. Notably, each HK+ mutant is associated with the activation of specific genes, except the global regulator VicRK (Fig. 2D and Supplementary Data 4E). As expected, the $VicK_{T221A}$ regulon included several operons involved in cell wall metabolism (Supplementary Fig. 3). However, constitutive activation of VicK probably leads to the activation of related stress and cell-wall signalling pathways. To identify relationships between TCS pathways involved in related processes, we analysed the 219 genes activated in at least one HK+ mutant for their expression in the whole RNA-seq dataset. This analysis confirmed the partial activation of SaeRS

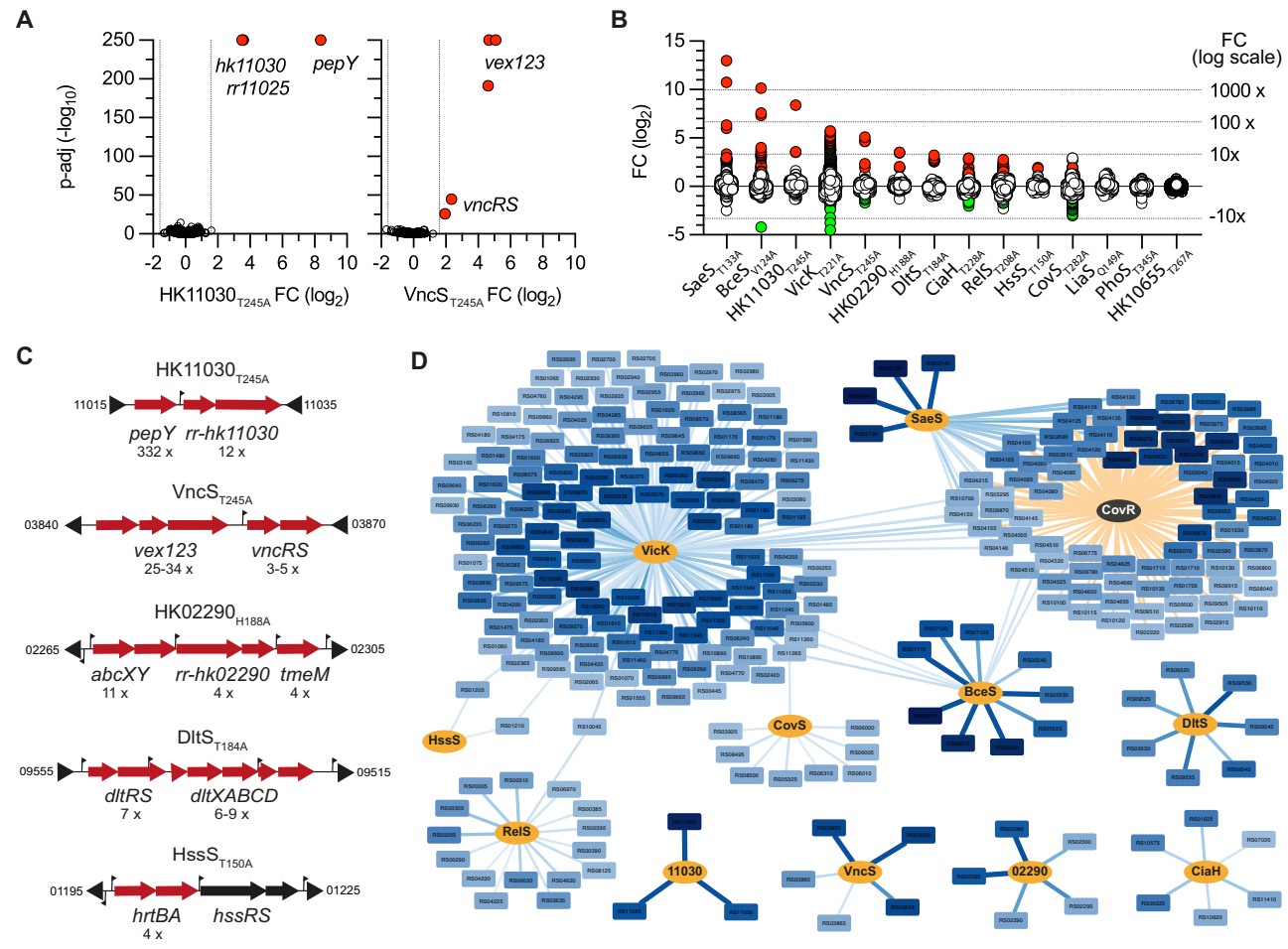

**Fig. 2 | The activated gene regulatory networks. A** Volcano plot of significant differential gene expression in the HK11030$_{T245A}$ (left panel) and VncS$_{T245A}$ (right panel) mutants. Transcriptomes by RNA-seq against the WT strain were done in the exponential growth phase in rich media (THY) with biological triplicates ($n$ = 3). Statistical analysis with DeSeq2 included Benjamini and Hochberg multiple comparisons to adjust $P$-values. Red dots highlight significantly differentially regulated genes above the thresholds FC > 3, p-adj <10$^{-4}$. Volcano plots for all mutants are provided in Supplementary Fig. 2. **B** Violin distribution of transcriptional fold change in the 14 HK$^+$ mutants. Coloured dots represent significantly activated (red) and repressed (green) genes (|FC| > 3, p-adj <10$^{-4}$), respectively. **C** Activated chromosomal loci in selected HK$^+$ mutants. Fold changes are indicated below the activated genes (red arrows). Transcriptional start sites identified by genome-wide TSS mapping are represented by vertical flags. NCBI gene ID bordering the loci are shown in a shortened form (e.g., 11015 = BQ8897_RS11015). Activated loci for each HK$^+$ mutant are provided in Supplementary Fig. 3. **D** Network of activated genes. Histidine kinases (orange nodes) are connected to their activated genes (light to dark blue nodes). Edge thickness and gene node colour are proportional to statistical significance and fold change, respectively. Activated genes in the CovR$_{D53A}$ mutant are included to account for the specificity of CovR as a global repressor (i.e., pale edges = CovR repression). Source Data for panels (**A**, **B**, **C**, and **D**) are provided in Supplementary Data 4.

signalling in the VicK$_{T221A}$ mutant (a shared CovR connection, in fact, see specific section below) sustained by genes with 1 < FC < 3 and significant but higher p-adj value compared to the SaeS$_{T133A}$ activated system (Supplementary Data 4H). Similarly, by considering significantly regulated genes with lower thresholds (1 < |FC| < 3, p-adj< 0.05), significant positive or antagonistic interactions were detected between signalling pathways (e.g., DltS activating CiaH and VicK, CiaH antagonising RelS, HK02290 antagonising HK11030). Finally, relaxing the thresholds also reveals the first five genes of the *phoRS* operon as the most and only significantly up-regulated genes (1,5 < FC < 1,75; 7.10$^{-3}$ < p-adj< 10$^{-5}$) in the PhoS$_{T345A}$ mutant (Supplementary Data 4D), suggesting a conserved mechanism of phosphatase activity but an inefficient activation of the PhoR regulator in the corresponding HK$^+$ mutant.

## Activation of the global repressor of virulence CovRS

The CovRS system is the major regulator of virulence in GBS and, in contrast to canonical TCS, acts as a global repressor of gene transcription[45]. Analysis of RNA-seq to identify negative regulation using similar thresholds (−3 > FC, p-adj < 10$^{-4}$) revealed the repression of 32 genes in the whole dataset (Supplementary Data 4G). Almost all repressed genes are regulated by the two global regulators, VicK (17 genes) and CovS (14 genes). Notably, the most highly repressed gene (22-fold in VicK$_{T221A}$) encodes a D-L endopeptidase, highlighting the conservation of negative regulation of cell wall hydrolases between WalRK homologous systems[41].

On the other hand, a comparative analysis of repressed genes in CovS$_{T282A}$ with the known CovR regulon shows the limitations of the HK$^+$ approach for characterising the CovRS system. Indeed, only 5 out of 14 repressed genes in the CovS$_{T282A}$ mutant belong to the CovR regulon of 153 genes previously determined with loss of function mutants[45]. This difference can be attributed to another specific feature of the CovRS system, which is active in the absence of an environmental signal[45]. The CovS$_{T282A}$ transcriptome, therefore, suggests that CovR over-activation does not translate into increased repression of targeted genes.

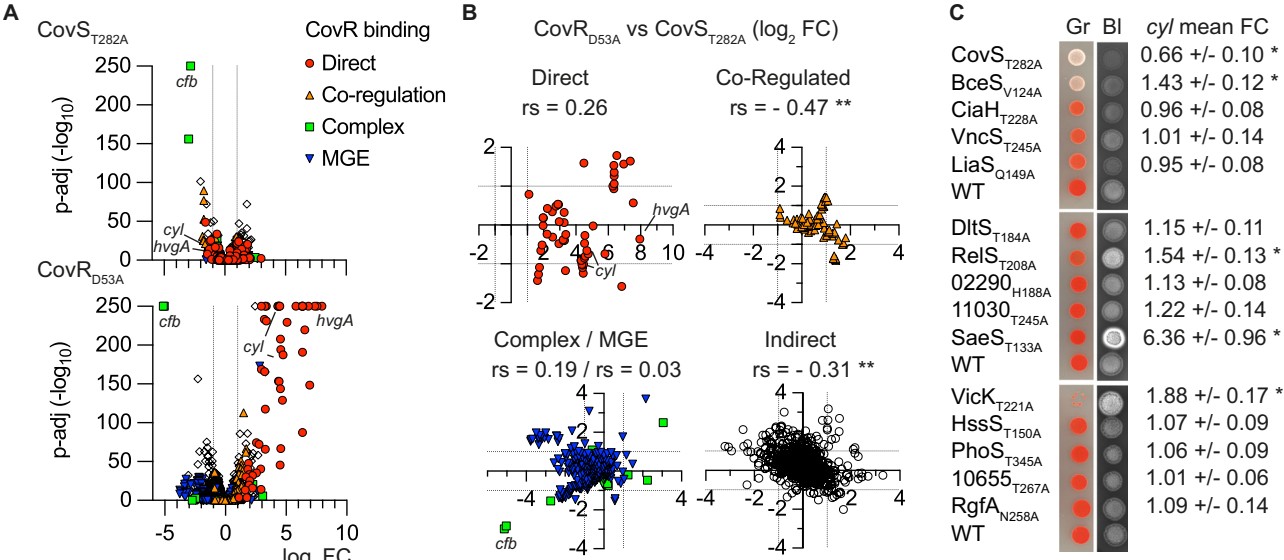

**Fig. 3 | Activation of the global repressor of virulence CovR. A** Volcano plot of significant fold changes in the CovR-active (CovS_{T282A}) and CovR-inactive (CovR_{D53A}) mutants. RNA-seq analyses with biological triplicate ($n = 3$) were analysed with Benjamini and Hochberg multiple comparisons to adjust $P$-values. Coloured dots highlight genes according to CovR-regulatory mechanisms as previously defined by genome-wide binding: direct repression (red circles), CovR-binding requiring additional regulators for activation (orange triangle), atypical CovR-binding inside ORFs or positive regulation (green square), silencing or anti-silencing of genes in mobile genetic elements (inverted blue triangle). **B** Comparison of fold changes between CovR inactivation and activation according to regulatory mechanisms. Significant correlations are highlighted (**: rs non-parametric Spearman correlation, $p$ (one-tailed) $< 10^{-4}$). **C** Pigmentation and haemolytic phenotypes of the HK⁺ mutants on selective media. Spots of diluted cultures are incubated on Granada (Gr) and Columbia horse blood (Bl) plates in anaerobiosis and aerobiosis, respectively. Haemolytic activity is visualised by the dark halo on the inverted black-and-white photographs. The BM110 parental strain (WT) was added to each plate as a control. Note that the VicK_{T221A} mutant does not grow on Granada media, the basis of this phenotype requiring further investigation. The mean RNA-seq fold change with SD of the 12 genes $cyl$ operon encoding the pigmented haemolysin ß-h/c directly repressed by CovR is indicated, and statistical significance after Benjamini and Hochberg multiple comparison is highlighted (* p-adj < 0.005). Source data for all panels are provided as a Source Data file.

To further analyse the CovRS system, we compared the transcriptome of the activated CovS_{T282A} mutant with that of the inactivated CovR_{D53A} mutant, which cannot be phosphorylated by CovS, and also included our genome-wide CovR binding analysis done by ChIP-seq[45]. Side-by-side transcriptome comparison showed that the inactivation of CovR activates the signalling pathway repressed by the active CovRS system, without a general inverse relationship in the CovS_{T282A} mutant (Fig. 3A). Nevertheless, a significant inverse correlation between the CovS_{T282A} and CovR_{D53A} transcriptomes is observed for genes that do not belong to the direct CovR regulon (Fig. 3B), suggesting that overactivation of CovR primarily increases binding to co-regulated promoters and low-affinity binding sites[45].

Since the inactivation of the CovRS repressor system is more informative than its over-activation, we included the CovR_{D53A} transcriptome in the HK⁺ dataset. The global gene network reveals connections between CovR-repressed genes and SaeS_{T133A}, VicK_{T221A}, or BceS_{V124A} activated genes (Fig. 2D). Notably, SaeS_{T133A} is highly connected with the direct CovR-repressed genes, while BceS_{V124A} activates only three CovR indirectly regulated genes. Functional assays using pigmented beta-haemolysin/cytolysin (ß-h/c) production as a natural reporter of CovR activity first confirmed the non-pigmented and non-haemolytic phenotypes of the CovS_{T282A} mutant (Fig. 3C), in agreement with CovR directly repressing the $cyl$ operon encoding the ß-h/c synthesis and export machineries[45,46]. The phenotype of six additional HK⁺ mutants are different from the WT strain on selective media, either increasing (SaeS_{T133A}, VicK_{T221A}, and RelS_{T208A}) or decreasing (BceS_{V124A}, CiaH_{T228A}, and LiaS_{Q149A}) pigmentation and/or haemolytic activity (Fig. 3C). However, the absence of correlation between the transcription of the $cyl$ operon and the pigmentation/haemolytic phenotypes in several HK⁺ mutants (Fig. 3C) suggests that ß-h/c activity depends on post-transcriptional regulatory mechanisms in addition to CovR regulation of the $cyl$ operon. This is potentially the case in mutants with altered cell surface composition, where the toxin interaction occurs[47,48].

## The PbsP adhesin connects SaeRS and CovRS signalling

We sought to decipher the connection between the CovRS and SaeRS systems, two main regulators of host-pathogen interactions. Published transcriptomes with $saeRS$ deletion mutants define a large regulon of 400–600 genes depending on growth conditions[49]. In contrast, analysis of the SaeS_{T133A} transcriptome reveals the highly significant ($60 < FC < 8000$-fold, p-adj $< 10^{-250}$) activation of four genes only, along with a partial activation of the CovR regulon (Fig. 4A). We confirmed the stratification of the SaeS_{T133A} differentially regulated genes by RT-qPCR, validating 3 groups: the $pbsP$ and $bvaP$ genes, the $saeRS$ operon, and the CovR-regulated genes represented by the directly repressed genes $cylE$ and $hvgA$ (Fig. 4B).

Intrigued by the almost 50-fold difference between $pbsP$ and $saeRS$ up-regulation, we analysed the genomic locus in detail. The 112 bp $pbsP$-$saeRS$ intergenic region contains a $P_{saeR}$ promoter but no canonical transcriptional terminator. The integration of such a terminator precisely after the $pbsP$ stop codon in the SaeS_{T133A} mutant abolishes $saeRS$ overexpression while having no impact on other activated genes (Fig. 4B). Quantification of promoter activities using ß-galactosidase reporters confirms a similar activity of $P_{saeR}$ in the WT and in SaeS_{T133A} mutant, and the strong activation of $P_{pbsP}$ upon activation of the SaeRS system (Fig. 4C). This shows an indirect positive feedback loop of the $saeRS$ operon, which is transcribed by its constitutive promoter and regulated by $pbsP$ termination readthrough. Interestingly, the basal level of $saeRS$ transcription in SaeS_{T133A} with the $pbsP$ terminator is sufficient to fully activate $pbsP$ and $bvaP$ (Fig. 4B), implying that the indirect feedback loop may be physiologically relevant only for controlling the kinetics of the signal-dependent response, but not its amplitude.

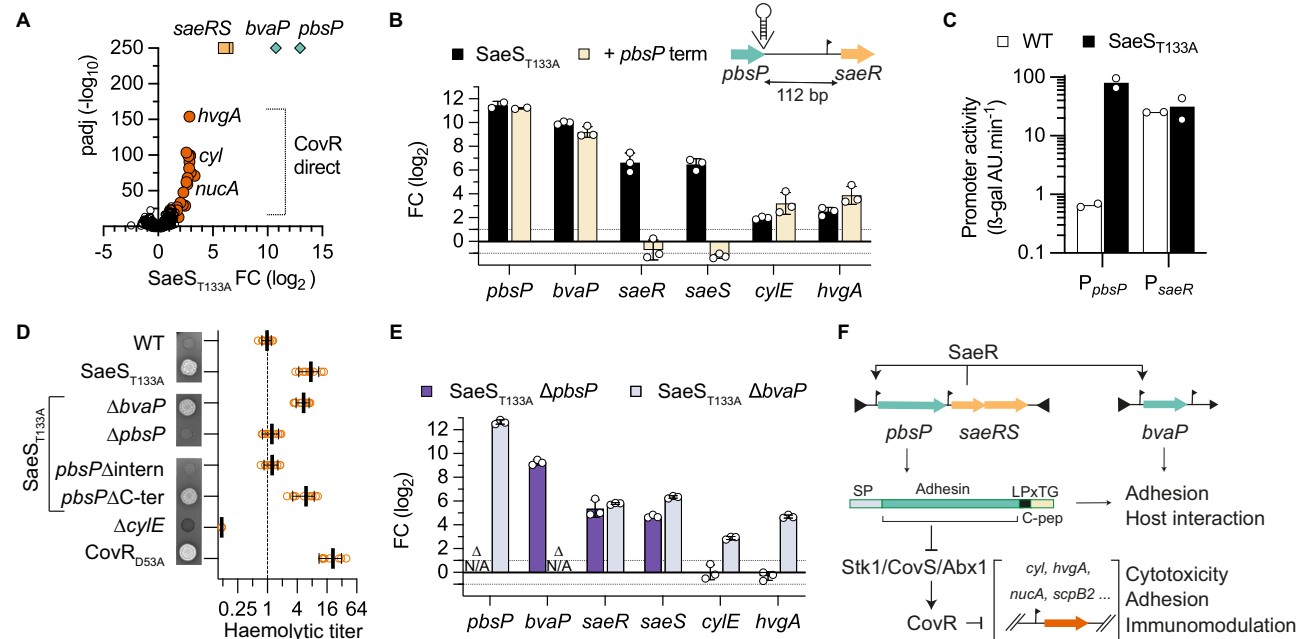

**Fig. 4 | Adhesin-dependent wiring of the SaeR and CovR regulatory networks.**
**A** Volcano plot of significant fold change in the SaeS$_{T133A}$ mutant. RNA-seq analyses with biological triplicate ($n = 3$) were analysed with Benjamini and Hochberg multiple comparisons to adjust $P$-values. Dot colours highlight the stratification of activated genes between *pbsP* and *bvaP* (green), the *saeRS* operon (orange) and the CovR-regulated (red) genes. **B** Indirect positive feedback loop of the *saeRS* operon. The *pbsP* and *saeRS* genes are separated by a 112 bp intergenic region containing a *saeR* transcriptional start site located 31 bp from the SaeR start codon. Transcriptional activation of *pbsP* and *saeRS* is uncoupled by the integration of a canonical terminator at the 3′ end of *pbsP*. Fold changes of selected genes are quantified by RT-qPCR in the SaeS$_{T133A}$ (black bars) and in the SaeS$_{T133A}$ + *pbsP* terminator (light bars) mutants. Bars represent the mean with SD of biological replicate ($n = 3$). **C** Activities of the P$_{pbsP}$ and P$_{saeR}$ promoters in the WT and SaeS$_{T133A}$ mutant. Bars represent the activity of the ectopic ß-galactosidase reporter system under the control of the tested promoters in the WT (white bars) and the SaeS$_{T133A}$ (black bars) mutant. Bars represent the mean of two biological duplicates ($n = 2$). **D** Hyper-haemolytic activity of the SaeS$_{T133A}$ mutant is dependent on the PbsP adhesin. Qualitative and semi-quantitative haemolytic activity is tested on Columbia blood agar media and with defibrinated horse blood, respectively. The $\Delta cylE$ and CovR$_{D53A}$ mutants are included as negative and positive controls, respectively. Haemolytic titres are normalised against the WT strain. Individual data points from biological replicates ($n = 10$, except for WT $n = 9$, CovR$_{D53A}$ $n = 8$, and $\Delta cylE$ $n = 4$) are represented with their mean +/− SD. **E** Upregulation of the PbsP adhesin activates CovR-regulated genes. Transcriptional fold change of selected genes by RT-qPCR in the SaeS$_{T133A}$ $\Delta pbsP$ (light blue) and SaeS$_{T133A}$ $\Delta bvaP$ (dark blue) double mutants. Bars represent the mean with SD of biological replicate ($n = 3$). **F** Wiring diagram of the SaeRS signalling pathway. The *saeRS* operon is transcribed at a basal level by a constitutive promoter. Upon TCS activation, the SaeR regulator activates the transcription of genes encoding the PbsP and BvaP virulence factors and indirectly its own operon through a *pbsP* terminator readthrough. The overexpression of the PbsP adhesin domain, but not the carboxy-terminal part containing the LPxTG anchoring motif and the hydrophobic C-peptide, is necessary to trigger CovR-regulated virulence factor expression via the Stk1/CovS/Abx1 regulatory proteins. Source data for panel (**A**) are provided in Supplementary Data 4 and as a Source Data file for panels (**B**, **C**, **D**, and **E**).

We next analysed the connection between SaeRS and CovRS signalling. The activation of CovR-regulated genes in the SaeS$_{T133A}$ mutant is intermediate when compared to the CovR$_{D53A}$ mutant (Figs. 3A, 4A). One hypothesis could be a competitive binding between SaeR and CovR, but it is unlikely that all binding sites will allow both SaeR-activation and CovR-repression. As an alternative, we hypothesised that the two genes specifically regulated in the SaeS$_{T133A}$ mutant, encoding the PbsP cell-wall anchored adhesin[50,51] and the BvaP secreted protein[52], could be involved in the activation of the CovR regulon. Indeed, the deletion of *pbsP*, but not of *bvaP*, in the SaeS$_{T133A}$ mutant, restores a WT haemolytic activity (Fig. 4D). In agreement with the phenotypes, the deletion of *pbsP* in the SaeS$_{T133A}$ mutant restores a WT level of the CovR-regulated genes *cylE* and *hvgA*, while the *saeRS* and *bvaP* genes are still similarly up-regulated (Fig. 4E).

After cleavage by the enzyme sortase A and anchoring to the cell wall, the remaining carboxy-terminal domain of an LPxTG adhesin can act as a signalling molecule by interacting with the transmembrane domain of a specific HK, as demonstrated in *Streptococcus gordonii*[53]. We, therefore, considered this C-peptide mechanism and constructed mutants expressing truncated PbsP variants in the SaeS$_{T133A}$ mutant. In-frame deletion of the PbsP C-peptide (e.g., 108 bp deletion including the LPxTG cell-wall anchoring motif until the penultimate codon) has no effect on the induction of the CovR-regulated haemolytic activity

(Fig. 4D). In contrast, in-frame deletion of the PbsP adhesin domain (1239 bp deletion leaving the signal peptide and the LPxTG cell wall anchoring motif intact) restores the haemolytic activity of the SaeS$_{T133A}$ mutant to WT level (Fig. 4D). Furthermore, the growth defect of the SaeS$_{T133A}$ mutant, which is similar to the growth defect of the CovR$_{D53A}$ mutant, is suppressed by deletion of *pbsP* or of the adhesin part of *pbsP* (Supplementary Fig. 4). Thus, the PbsP adhesin domain triggers CovR signalling either by interacting with CovS or co-regulatory proteins[54] or by inducing surface perturbations specifically sensed by the CovRS system (Fig. 4F).

## Drug-independent activity of the BceRS three-component system

We next sought to decipher the function of the BceRS system, which, upon activation, shows hypo-pigmented and haemolytic phenotypes that are independent of CovRS regulation of the *cyl* operon (Fig. 3C). The BceRS system belongs to a conserved TCS family that relies on a transporter to sense and transmit environmental signals to the HK[55]. The transcriptome of the BceS$_{V124A}$ mutant reveals a 9-gene regulon, including the *bceRS* operon and adjacent genes (Fig. 5A and Supplementary Figs. 2 and 3). Further validation by RT-qPCR confirmed the 10- to 1000-fold activation of the regulon in the absence of drugs in the BceS$_{V124A}$ mutant, as well as the absence of significant transcriptional

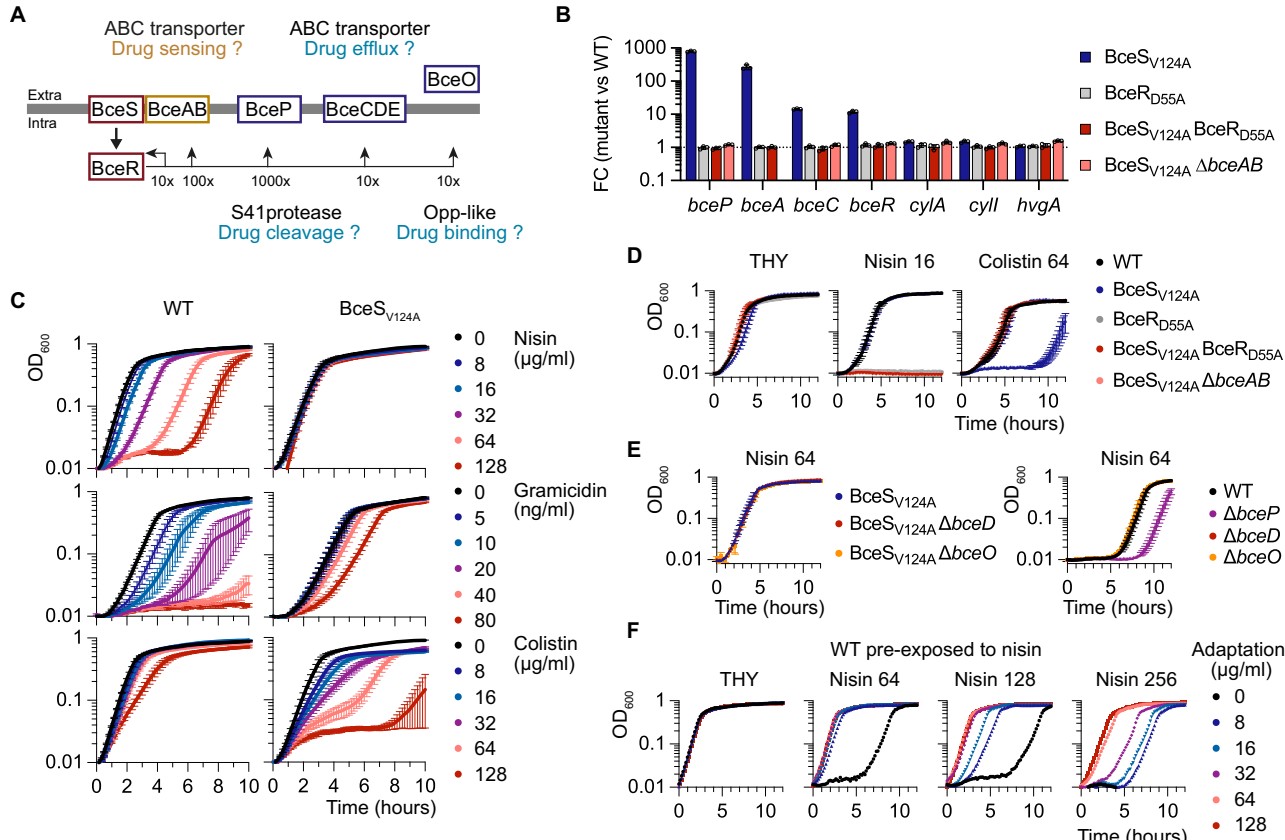

**Fig. 5 | The BceRS three-component system controls an adaptive response.**
**A** Schematic of the BceRS signalling pathway. The RNA-seq fold change scale in the BceS$_{V124A}$ mutant is shown below the horizontal line. The BceAB transporter is the third component of the regulatory system, predicted to sense and transduce the signal to BceS. Functions currently assigned to each component are indicated by question marks with coloured frames indicating functional category (red: signalling; yellow: sensing and transmission; blue: drug resistance). **B** BceAB is necessary to activate BceRS signalling in the absence of drugs. Fold changes during exponential growth in rich media were quantified by RT-qPCR in the activated HK$^+$ mutant (BceS$_{V124A}$: blue), in mutants with a non-phosphorylable variant of the cognate regulator in the WT (BceR$_{D55A}$: grey) or activated (BceS$_{V124A}$ BceR$_{D55A}$: red) backgrounds, and in a BceAB transporter mutant in the activated background (BceS$_{V124A}$ $\Delta bceAB$: pink). Bars represent the mean and SD of biological replicates ($n = 3$). **C** Growth curves of the WT and activated BceS$_{V124A}$ mutant in the presence of an increasing concentration of drugs. The curves represent the mean and SEM of biological replicates ($n = 4$). **D** Drug susceptibilities of double mutants abolishing BceRS activation in the BceS$_{V124A}$ mutant. The curves represent the mean and SEM of biological replicates ($n = 3$). **E** Drug susceptibilities of $\Delta bceP$, $\Delta bceD$, and $\Delta bceO$ mutants in the activated mutants (left panel) and/or the WT strain (right panel). The curves represent the mean and SEM of biological replicates ($n = 3$). **F** Growth curves of the WT strain pre-exposed to nisin. Early exponential growing WT strains were exposed for 4 h to nisin (Adapt 0 to 128 µg/ml) in THY at 37 °C. After washing and OD$_{600}$ normalisation, each culture is inoculated in fresh, rich media (THY) and with increasing concentrations of nisin (Nisin 64, 128, and 256 µg/ml). The curves represent the mean of two biological replicates ($n = 2$). Source data for all panels are provided as a Source Data file.

changes of the CovR-regulated genes (Fig. 5B). As expected, mutation of the BceR regulator to a non-phosphorylated form (BceR$_{D55A}$) abolishes the activation of the signalling pathway in the BceS$_{V124A}$ mutant (Fig. 5B). Interestingly, deletion of the transporter/sensor ($\Delta bceAB$) in the BceS$_{V124A}$ mutant also switches off the signalling pathway (Fig. 5B), showing the essential role of the BceAB transporter in activating BceRS signalling in the absence of inducing signals.

Typically, this TCS family confers resistance to antimicrobials targeting lipid II cell wall metabolites such as nisin or bacitracin. Genetic activation of the pathway renders the BceS$_{V124A}$ mutant insensitive to nisin, which has a marked effect on the lag phase but not on the growth rate of the WT strain, and increases resistance to gramicidin and, to a lesser extent, bacitracin (Fig. 5C and Supplementary Fig. 5). Interestingly, the BceS$_{V124A}$ mutant is also more susceptible to antimicrobial peptides (colistin and polymyxin D) compared to the WT strain, while equally susceptible as the WT to vancomycin (Fig. 5C and Supplementary Fig. 5). These results show that BceRS activation confers protection against structurally unrelated drugs targeting lipid II intermediates at the cost of increased susceptibility to antimicrobial peptides.

We then tested the phenotypes associated with loss-of-function of BceRS. Inactivation of the pathway in the BceS$_{V124A}$ background by additional BceR$_{D55A}$ or $\Delta bceAB$ mutations leads to nisin hyper-susceptibility while restoring WT level of colistin susceptibility (Fig. 5D). Nisin hyper-susceptibility is also observed for the single BceR$_{D55A}$ mutant (Fig. 5D), a phenotype not linked to down expression of BceRS regulated genes (Fig. 5B). These results show that the BceRS system is constitutively active in the absence of drugs and that the basal activity is necessary and sufficient to counteract the effects of sub-inhibitory concentrations of nisin.

To test the current model of nisin resistance based on drug efflux and cleavage, we inactivated the BceCDE transporter, the BceO substrate-binding protein, and the BceP extracellular protease (Fig. 5A). Deletion of $\Delta bceD$ and $\Delta bceO$ in the WT or BceS$_{V124A}$ backgrounds has no impact on the nisin phenotypes of the respective parental strains (Fig. 5E), excluding a major function in drug export or binding. In contrast, the $\Delta bceP$ mutant is slightly more susceptible to nisin compared to the WT parental strain (Fig. 5E). However, deletion of $bceP$ in the BceS$_{V124A}$ background was always associated with secondary mutations inactivating the whole signalling pathway in five

independent mutants (Supplementary Data 2). Altogether, these results show that individual genes do not provide drug resistance and suggest that the BceP extracellular S41 protease[56] has a buffering role when the pathway is activated, rather than directly cleaving drugs through an atypical mechanism, as previously suggested[57].

To test if BceRS regulates an adaptive response rather than a resistance mechanism per se, we pre-incubated the WT strain with nisin for four hours. Prior exposure to the drug decreases the lag phase in a dose-dependent manner upon subsequent exposure to higher nisin concentrations (Fig. 5F). For instance, adaptation with 8 μg/ml nisin, a WT sub-inhibitory concentration, confers a BceS$_{V124A}$-like resistance against a subsequent 64 μg/ml nisin challenge (Fig. 5F). More generally, prior adaptation with a given nisin concentration increases by a 4-fold factor the inhibiting concentration. These results show that the BceRS response is adaptive and suggests that the BceRS system actively monitors and adjusts surface-exposed lipid II metabolites, rather than directly detoxifying drugs or drug-lipid II complexes.

## Discussion

Our systematic analysis highlights the benefits of the HK$^+$ gain-of-function approach to characterise TCS signalling, both for mapping regulatory networks and for characterising individual systems. This study was made possible by the conserved mechanism of HK phosphatase activity originally proposed[31,32], which allowed a single residue to be targeted to activate the corresponding signalling pathway. By systematically testing all HisKA and HisKA_3 systems in a bacterium, we show the broad potential of this approach to reveal specialised, connected, and global regulatory systems covering the functional diversity that has evolved from a simple two-component architecture.

Targeting the HK phosphatase catalytic residue has the advantage of leaving a quasi-native system. The gain-of-function is solely dependent on the HK mutation, with no change to the RR and preservation of the physiological feedback loops. A second major advantage is that it bypasses the need for environmental signals, which are often unknown or confounding when having a wide effect on bacterial physiology. In this respect, the SaeRS system is a remarkable example. Previous studies demonstrated SaeR regulation of *pbsP* and *bvaP* during vaginal colonisation, among transcriptomic perturbations affecting nearly 40% of the genome[49]. However, the regulon remained elusive due to a lack of activation in vitro[49]. The HK$^+$ approach resolves the signalling pathway by revealing a specialised and CovR-connected, pathway. Comparison with the well-characterised *Staphylococcus aureus* homologous system[58] highlights the evolutionary divergence between regulatory circuits, particularly for those regulating host-pathogen interactions, which need to be studied in each species.

Originality is a mechanism linking the SaeRS and CovRS systems. Complex regulatory wiring can be selected to mount co-ordinated responses, primary trough transcriptional cascades (a TCS regulating transcription of a second TCS) or connectors (usually a TCS-regulated transmembrane protein modulating the activity of a second TCS)[59]. The C-peptide of adhesins can act as connectors when the transmembrane end remaining after cleavage of the LPxTG motif by sortase A interacts with a histidine kinase[53]. The mechanism differs in GBS in which the PbsP adhesin domain acts as an extracellular signalling molecule to activate CovR signalling, independently of cell wall anchoring. We hypothesise that the lysin-rich and positively charged PbsP adhesin interacts with CovS, with the co-regulatory proteins Abx1 and Stk1[54,60], or with the negatively charged membrane, recalling the activation of the homologous CovRS system in *Streptococcus pyogenes* by cationic peptides[61]. To complete the regulatory circuit, CovR has previously been shown to repress *pbsP* in a strain-specific manner[45,50,51]. The intertwining of SaeR and CovR signalling through PbsP constitutes an adaptive mechanism for balancing adhesion and invasion and could

contribute to the phenotypic and pathogenicity variabilities observed within the species.

HK$^+$ mutations resolve TCS regulatory networks but reveal discrepancies in the activation of signalling pathways. While primary sequence analysis of TCSs did not uncover specific motifs correlating with high, intermediate, or low pathway activation, two underlying factors may dampen the effect of HK$^+$ mutations. First, HK kinase activity can be inhibited by interacting proteins, such as the small LiaF protein inhibiting LiaS[62,63] and the Pst/PhoU proteins inhibiting PhoRS[64,65]. Genes encoding co-regulatory proteins are often themselves regulated by the TCS, creating feedback loops that lock HKs in kinase-deficient conformation and thus obliterate the effect of HK$^+$ mutations. However, the presence of auto-inhibitory proteins is not a sufficient condition for preventing activation, as demonstrated by VicK, which is inhibited by YycH/YycI[66,67] but still activated by an HK$^+$ mutation. Second, intermediary activation of the RelRS and CiaRH pathways suggests buffering mechanisms for TCSs regulating multiple independent loci and integrated cellular response. However, detailed analysis is required to decipher phosphorylation dynamics in each phosphatase-deficient HK$^+$ and correlate in vivo RR phosphorylation with regulatory network activation, considering variable factors like the source of RR phosphorylation (kinase activity of the HK$^+$ variant, cross-talk by other HK, small metabolites) and specific spontaneous dephosphorylation rates of the labile aspartate residues[27,68,69].

The systematic approach validates the conservation of the dephosphorylation mechanism. It also uncovers an unanticipated activation of the BceRS system with a degenerate QMKV motif. Recent structural insights from *Bacillus subtilis* complexes into the membrane environment support a highly dynamic model of interactions between the BceAB transporter and the BceS kinase- and phosphatase-competent conformations[70,71]. Our results with the HK$^+$ BceS indeed suggest that BceAB is necessary to stabilise the kinase-competent conformation of BceS. Alternatively, BceAB could also provide the catalytic residue on the models suggested for the auxiliary phosphatases RapH and Spo0E[31,72]. At the phenotypic level, our results point towards a need-based mechanism of target protection, as recently suggested for the Bce-like system[73–75], and not towards a drug cleavage-exclusion mechanism as initially suggested[76,77]. The target protection mechanism relies on the binding of lipid II intermediates on a binding pocket of BceAB[70]. However, it is still unclear how the system releases lipid II when it is complexed with drugs. Our results suggest an alternative scenario in which BceAB constantly monitor free lipid II intermediate to minimise target exposure[78,79]. This alternative is supported by the steady-state activity of the BceRS pathway in the absence of drugs and is compatible with a need-based mechanism. Further studies should test the entire BceRS pathway without relying on a lipid II-drug detoxification mechanism but rather on a mechanism that maintains the steady-state level of free lipid II in the presence of drugs. It is also interesting to note that the activation of the BceRS-BceAB system impacts the retention or secretion of the ß-h/c toxin, potentially as a result of the interaction between the polyene backbone of the toxin and the membranes[46–48], suggesting functional links between cell envelope homoeostasis, drug resistance, and virulence.

To conclude, genetic activation by HK$^+$ is a powerful approach to characterise positive regulation by TCS. It circumvents the major drawback of studying systems that are usually non-activated in standard conditions. Previous studies on individual TCSs have demonstrated the potential of the approach, but it has unfortunately not been widely adopted to date. Our systematic analysis based on the conserved mechanism of phosphatase activity provides a blueprint to decipher signalling, response dynamics, evolution of gene regulation, and regulatory networks. The HK$^+$ approach is recommended for the study of TCS in any species, either as a complement or as a first choice alongside a classical deletion mutant.

## Methods

### Strain, bacterial genetics, and genome sequencing

The BM110 strain is a clinical isolate representative of the hypervirulent CC-17 clonal complex responsible for most neonatal meningitis[80]. The 2.2 Mb annotated genome is available under the NCBI RefSeq reference NZ_LT714196. The standard growth condition is in Todd-Hewitt medium supplemented with 1% yeast extract and 50 mM Hepes pH 7.4 (THY) incubated in static condition at 37 °C.

Oligonucleotides and construction of vectors for site-directed mutagenesis and deletion are detailed in Supplementary Datas 5 and 6, respectively. Splicing-by-overlap PCR with high-fidelity polymerase (Thermo Scientific Phusion Plus) were done with complementary primers containing the desired mutations. The final PCR products contain mutations (SNP or deletion) flanked on either side by 500 bp of sequence homologous to the targeted loci. Cloning is done by Gibson assembly in pG1, a thermosensitive shuttle vector similar to the pG+host5 vector[81]. Constructs were introduced and maintained in *E. coli* XL1-blue (Stratagene) with erythromycin selection (150 µg/ml) at 37 °C. Inserts were validated by Sanger sequencing (Eurofins Genomics).

Mutant construction in GBS was performed through a three-step process involving episomal replication, chromosomal integration, and vector loss. Initially, pG vectors were introduced into GBS via electroporation, and transformants were selected on THY agar supplemented with 5 µg/ml erythromycin at 30 °C, the permissive temperature for episomal replication. After 24–36 h of growth, two single transformant colonies were isolated on THY agar containing erythromycin and incubated another 24–36 h at 37 °C, the non-permissive temperature for vector replication, thereby promoting chromosomal integration of the vector at the targeted locus through homologous recombination. Subsequently, rare colonies that had integrated the vector into the chromosome were isolated on THY agar with erythromycin at 37 °C for 16 h. The isolated integrants were inoculated into 10 ml of THY medium without antibiotic, incubated at 30 °C to activate the rolling circle origin of replication, and subjected to serial subculture twice daily. By day 3, cultures were diluted (typically $10^{-5}$), spread onto THY and Columbia agar supplemented with 10% horse blood (BioMerieux), and incubated at 37 °C. Isolated colonies ($n = 24$–48) were picked into 150 µl of THY in 96-well plates, incubated for 4–6 h at 37 °C, and replica-plated using a 96-pin replicator (Boekel Scientific) onto THY agar plates with and without erythromycin. After 16 h at 37 °C, erythromycin-susceptible colonies that had lost the vector were identified and confirmed by discriminatory PCR (MyTaq HS - Bioline) using specific oligonucleotides with the expected mutation at their 3' extremity (Supplementary Data 5) to distinguish mutant from wild-type genotypes. Mutants were then isolated on THY at 37 °C, single colonies inoculated in THY incubated at 37 °C for 16 h, centrifuged and resuspended in 20% glycerol for long-term storage at −80 °C.

Genomic DNA of at least two independent mutants for each construction were purified from 1 ml of culture following manufacturer instruction for Gram-positive bacteria (DNeasy Blood and Tissue – Qiagen) and sequenced (Illumina sequencing at Core facility or Eurofins Genomics). High-quality reads in Fastq were mapped against the BM110 genomes and analysed with Geneious Prime (2019.2.3 – Biomatters Ltd) using default parameters (mapping: up to 5 iterations, minimum mapping quality Phred score 30, medium/low sensitivity; SNP calling: minimum variant frequency 0.25, maximum variant $P$-value $10^{-6}$, minimum strand-bias $P$-value $10^{-5}$, with visual inspection for coverage and validation of SNPs). Results of genome sequencing for all mutants used in this study (55–419 x coverage, mean 181 x) are summarised in Supplementary Data 2.

### RNA sequencing

RNA purification, sequencing and analysis were conducted essentially as described for the characterisation of the virulence regulator CovR[45].

The 14 HK$^+$ mutants were split into two series of 8 strains (7 mutants and one WT strain), and RNA was purified using three independent replicates that were grown on different days. Overnight cultures were used to inoculate THY (1/50), and 10 ml of culture were harvested in the exponential growth phase ($OD_{600} = 0.5$) after incubation at 37 °C. Bacterial pellets are washed with cold PBS containing RNA stabilisation reagents (RNAprotect, Qiagen) before flash freezing and storage at −80 °C. Total RNA are extracted after cell wall mechanical lysis with 0.1 µm beads (Precellys Evolution, Bertin Technologies) in RNApro reagent (MP Biomedicals), and purified by chloroform extraction and ethanol precipitation.

Samples were treated to remove residual DNA (TURBO DNase, Ambion) before fluorescent-based quantification (Qubit RNA HS, Invitrogen) and quality validation (Agilent Bioanalyzer 2100). Depletion of rRNA (FastSelect Bacterial, Qiagen), library construction and sequencing were done following manufacturer instructions (TruSeq Stranded mRNA, NextSeq 500, Illumina). Single-end strand-specific 75 bp reads were cleaned (cutadapt v2.10) and mapped on the BM110 genome (Bowtie v2.5.1, with default parameters). Gene counts (featureCounts, v2.0.0, parameters: -t gene -g locus_tag -s 1) were analysed with R (v4.0.5) and the Bioconductor package DESeq2 (v1.30.1)[82]. Normalisation, dispersion, and statistical tests for differential expression were performed with independent filtering. For each comparison, raw $p$-values were adjusted using Benjamini and Hochberg multiple tests[83] and adjusted $p$-values lower than 0.05 were considered significant. Raw sequencing reads and statistical analysis are publicly available (GEO accession number GSE261394).

In addition to HK$^+$ RNA-sequencing, we have included an independent CovS$_{T282A}$ transcriptome that was done simultaneously with the CovR$_{D53A}$ transcriptome[45], the latter being already reported altogether with CovR ChIP-seq experiment (GEO accession number GSE158049). Gene networks are represented with the open-source software Cytoscape (v3.9.1)[84].

### RT-qPCR and promoter activity

For validation, independent RNA purifications from biological triplicates were done using the same protocol, except that the cultures were grown on the same day and only 1 ml was harvested. Reverse transcription and quantitative PCR (iScript Reverse Transcription and SsoAdvanced Universal SYBR Green, BioRad) were done using specific primers (Supplementary Data 5). Fold changes are calculated for each target relative to the WT strain whose RNA was purified in parallel.

For promoter activities, promoters were amplified and cloned in the pTCV-lac vector containing a ß-galactosidase reporter (Supplementary Datas 5 and 6) and introduced in GBS. Reporter activity was quantified in microplate format by colourimetric assay with ONPG as substrate and permeabilized overnight cultures[54]. Reaction kinetics at 28 °C were followed by OD at 420 nm every 5 min (Tecan Infinite). Linear slopes (OD/min) were used to infer enzymatic activities and were normalised for the initial cell density (OD 600 nm) of each replicate.

### Growth curves and antibiotic susceptibilities

Growth curves are done in a volume of 150 µl of THY inoculated with diluted overnight cultures (1/500) in 96-wells microplates and incubated at 37 °C with automatic recording of OD 600 nm every 10 minutes and 1 min agitation by cycle (TECAN Infinite). Doubling times are determined by fitting non-linear regression with a Malthusian growth model (GraphPad Prism 10) in the exponential phase ($R^2 > 0.99$) for each replicate. Fitness is calculated by dividing the mean doubling time of the WT by the doubling time of the mutant. For antibiotic susceptibilities, concentrated drugs (10 x) were added to an aliquot of the starting cultures and serial two-fold dilutions were done in the starting culture without drugs before incubation in the microplate reader. Minimal Inhibitory Concentration (MIC) is done following EUCAST

guidelines in Muller-Hinton Fastidious culture media (MH-F, Becton Dickinson) media using custom AST Sensititre 96 wells plates (ThermoScientific) and 18 h of incubation at 37 °C.

## ß-haemolytic activity

Columbia agar supplemented with 5% horse blood and Granada medium (BioMerieux) were used to visualise ß-haemolytic activity and pigmentation, respectively. Serial ten-fold dilutions of cultures were spotted onto media plates and then incubated under anaerobic conditions (AnaeroGen, Oxoid) at 37 °C. To highlight the halo of lysis around colonies, images are converted to greyscale and uniformly processed (Photoshop, Adobe) to adjust contrast and brightness. Haemolytic titres were determined by a semi-quantitative method[54]. Serial 2-fold dilution of cultures initially adjusted to $10^9$ CFU/ml in PBS were added (V/V) to 1% defibrinated horse blood (Oxoid) in PBS supplemented with 0.2% glucose. After 1 hour of incubation at 37 °C, cells were gently pelleted, and haemoglobin in supernatants was quantified by optical absorbance at 420 nm. The haemolytic activity of each strain was defined as the minimum dilution that lysed at least 50% of red blood cells. Haemolytic titres are the ratio between the haemolytic activity of each replicate against the haemolytic activity of the WT strain. Haemolytic titres are then normalised against the WT strain (normalised WT titre = 1).

## RR phosphorylation level

Genes encoding RR were amplified and cloned by Gibson assembly (Supplementary Data 5 and 6) in a custom-made pEX-CterFLAG vector containing a synthetic cassette with a translational initiation site, a flexible Gly-Ala linker, a 3xFLAG epitope, and a transcriptional terminator. Cassettes with genes of interest cloned in frame with the linker were excised with restriction enzymes and cloned into the anhydrotetracycline (aTc) inducible expression vector pTCV_P$_{teto}$[45]. Expression vectors were introduced in the corresponding HK$^+$ mutants by electroporation with kanamycin selection. Total protein extracts were prepared from 45 ml of cultures in exponential phase in the presence of 100 ng/ml aTc (Sigma) by mechanical lysis of bacterial pellet (Precellys Evolution) resuspend in cold TBS buffer with EDTA-free protease inhibitors (cOmplete, Roche). Following clearance by centrifugation, 15 µg of proteins were loaded in 12.5% Phos-Tag SDS polyacrylamide gels (SuperSep Phos-Tag, Wako Pure Chemical Industries Ltd) in loading dye buffer without EDTA and without sample heating to avoid dephosphorylation of the labile aspartate[85]. Electrophoresis (2 h, 100 V, 30 mA) in Tris-glycine buffer was performed on an ice bath. Semi-dry transfer on nitrocellulose membrane (15 min, 15 V, Mini-Protean, BioRad) was followed by blocking (TBS buffer with 0.05% Tween20 and 5% BSA), and hybridisation with rabbit polyclonal anti-FLAG antibodies (1:1500, Millipore F7425) and finally with secondary antibodies coupled to infra-red dyes (1:15000, Li-Cor 926-32211 IRDye 800CW). After final washing in TBS buffer without Tween20, fluorescent signals were acquired (Odyssey Imager, Li-Cor). The ratio of phosphorylated and non-phosphorylated proteins was analysed with ImageJ from three independent protein extracts.

## Reporting summary

Further information on research design is available in the Nature Portfolio Reporting Summary linked to this article.

## Data availability

The RNA-seq data generated in this study have been deposited in the Gene Expression Omnibus database under accession code GSE261394 [https://www.ncbi.nlm.nih.gov/geo/query/acc.cgi?acc=GSE261394].
Additional RNA-seq and ChIP-seq data already reported are available under accession code GSE261394 [https://www.ncbi.nlm.nih.gov/geo/query/acc.cgi?acc=GSE158049]. Source data are provided in this paper.

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

## Acknowledgements

This study was supported by Agence Nationale de la Recherche (VirEvol - ANR-22-CE15-0024) to A.F., and the National Laboratory of Excellence programme - Integrative Biology of Emerging Infectious Diseases (LabEx IBEID, ANR-10-LABX-62-IBEID) to P.T.C. C.C. and M.V.M. are recipients of National PhD grants from Ecole Doctorale BioSpc (ED562) - Université Paris Cité.

## Author contributions

C.C., F.C., M.V.M., C.G., O.S., and G.V.D.G. performed experiments and analysed data. E.J. and R.L. analysed RNA-seq data. G.T., P.T.C., A.T., C.B., and A.F. designed and supervised experiments. A.F. conceived the study and wrote the manuscript with input from all the authors.

## Competing interests

GT is an employee and CB is the founder and owner of Scylla Biotech Srl. The company did not provide funding and had no role in the design, conduct, or publication of the study. All other authors declare no competing interests.
