## [Peer Review File · Nature Communications]

Constitutive activation of two-component systems reveals regulatory network interactions in *Streptococcus agalactiae*Reviewer #1 (Remarks to the Author):

The study by Claverie et al. aims to conduct a systematic examination of the two-component sensory network (TCS) in *Streptococcus agalactiae*. The authors undertake a transcriptomic analysis of mutants in the phosphatase motif of each histidine kinase (HK⁺), assuming that this mutation results in constitutive phosphorylation of the RR. This assumption is necessary due to the experimental conditions, where the signals activating most TCSs are unlikely to be present. To validate their hypothesis, the authors assess the effect of HK⁺ on TCS expression and observe positive feedback in 7 HK⁺. Additionally, they express 3xFlag-tagged RRs in strains with wild-type HK and HK⁺, observing the phosphorylated form of RR in only two of 14 RRs via Western blot. Transcriptomic analysis reveals transcriptional profile changes in 10 of 12 TCS examined, with variable regulon sizes and overlap between regulons. From this point, the manuscript focuses on analysing specific aspects of the regulation of three specific TCS: CovRS, the role of adhesin PbsP in the regulation of SaeRS and CovRS, and the activity of BreRS. These three stories are very specific and unrelated, and it is difficult to understand how they contribute to the overall understanding of the *S. agalactiae* TCS network.

Major comments

Lines 110-116: In the case of WalkR, it is understood that compensatory mutations are generated when WalkR is mutated because its activity is essential, but in this case the mutation generated in VicST221A keeps the system constitutively active (phosphorylated) and the bacterium does not seem to like this and tries to compensate with other mutations. In general, all strains producing the mutant form of HK should be complemented with the wild-type form to exclude compensatory mutations in other genes.

Lines 134-136: To know if this induction is due to the vikS⁺ mutation and not to the off target mutation that the mutant has, the mutant would need to be complemented. RelRS pass the significance and the FC of 2? It seems that they are at the same point as vikRS in vikS⁺ in which they are not considered to be induced. Perhaps the significance of the TCS that pass it should be indicated in the graph with asterisks.

Line 168: Wouldn't there be 5? Hss, Rel, Sar and Bce but CovS also appears related to VikS. This interaction is shown in figure 2D but not mentioned in the text.

- Lines 206-208: The manuscript suddenly introduces a new mutant, CovRD53A, which cannot be phosphorylated. This sentence is not well understood. I think it means that inactivation of the RR activates the signalling pathways repressed by the active TCS. The way it is written, it seems very confusing to me. On the other hand, this comparison between the constitutively phosphorylated and non-phosphorylated form is very interesting and I think the manuscript should go in this direction rather than looking at specific aspects or phenotypes of CovRS and BceRS.

Lines 213-214 The authors admit that the analysis with the non-phosphorylatable mutant is more informative than with the constitutively active mutant. This statement seems to contradict their whole approach.

Line 220: The differences in haemolysis are not distinguishable in the photographs shown in the

figure.

Line 225-226: This phenotype seems to be multifactorial and not only related to the regulation of Cyl genes, I do not understand how this result facilitates the understanding of the message of the manuscript.

Minor comments

Line 118, reproducible, change for reproducibile

Line 119: “ and two having specific phenotypes” They would actually have a fitness disadvantage.

Lines 201-202: If the genes repressed by VicS are 17 and by Cov are 14, the total number of repressed genes is 31 and not 32.

Line 179: There is no section E in figure 2.

Line 220: The differences in haemolysis are not distinguishable in the photographs shown in the figure.

Figure 1C: HK+ mutants should be reported as SaeS+, etc. A consistent criterion in nomenclature should be maintained in all tables, figures and text. This would facilitate reading.

- Line 542: “...time ($F = Dx / DWT$ mean)” switch for “(Fitness = Doubling time x / Doubling time WT mean)”.

- Line 543: “(dashes)” switch for “(dashes lines)”.

Figure 2A: Indicate that 1000x, 100x, 10x refers to the FC.

The genes represented in the CovR node are induced in a CovR mutant, so they are repressed by CovR, while the rest of the nodes represent genes induced by the active form of HK (HK+). The genes of the CovR node should be shown in a different colour to distinguish repressed genes from induced genes.

Figure 4F: The figure should reflect what is mentioned in the text that the adhesin may be interacting with CovS or coregulatory proteins or inducing surface perturbations specifically detected by the CovRS system.

Supplementary Figure S2: HK11030 and VncS results are in Figure 2, remove them to avoid duplication.

Reviewer #2 (Remarks to the Author):

In this manuscript, the authors made phosphatase KO mutations in 14 sensor kinases of two-component systems in Group B strep and characterized their effect on growth, autoregulation, and cross-regulation. The key results are:

1. HK11050-RR11055 system and RelRS were cross-regulated by VikRS and CiaRH TCS, respectively.
2. The SaeRS TCS activates the CovRS TCS through the surface protein PbsP.
3. BceRS activation confers resistance to nisin and gramicidin but makes the cells more sensitive to colistin. Of the BceRS regulon, BceD and BceO were not required for the nisin

resistance, but BceP was required to some extent.

Based on these results, the authors concluded that the phosphatase KO mutation approach can be broadly applied to studying TCS in any species.

The methods employed are robust, and most data are of high quality and clear. In general, the authors' conclusions are well-supported by the presented data. The utility of the phosphatase KO approach was demonstrated for 11 TCSs. However, I still have some concerns:

1. It remains unclear whether the phosphatase KO mutation approach can be universally applied. Only two RRs out of 14 showed elevated phosphorylation in the phos-tag assay (Fig. 1D), and the HK+ mutation did not significantly affect the activity of LiaS and PhoS (Fig. S2).
2. The lack of a significant relationship between the expressions of direct targets of CovRS in CovR (inactive) and CovS (active) mutants is puzzling (Fig 3B). One would expect an inverse relationship, but this was not observed.
3. In Fig. 4D, it is necessary to confirm the proper expression of the C-terminal deletion mutant of PbsP (PbsPdC-ter).
4. The section from lines 317 to 328 is difficult to follow and understand. I do not see evidence supporting their claims such as “the BceRS system is active in the absence of drugs,” “The BceRS response is adaptative ..”, “... the response is constrained by its cost against antimicrobial peptides”, or “ the BCeRS system actively monitors and adjust surface-exposed lipid II metabolites.” It would be more suitable for the Discussion section rather than the Results.

Other minor comments

1. Please verify the correctness of the fitness formula (D_x/D_{wt}) in Fig 1B.
2. Line 119 “ .. two having specific phenotypes..”: It would be helpful if the authors specify the phenotypes.
3. Line 178-179, 181: There is no Fig. 2E. May be 2D?
4. Line 250: It is unclear what the authors mean by “dynamic” of the response.
5. Line 411: Hepes pH7.4 -> HEPES pH 7.4
6. Line 487: 18h -> 18 h
7. Line 504: I am not sure how Supplementary Table S4 is related to the cloning step.

Reviewer #3 (Remarks to the Author):

The manuscript of Claverie et al. describes the characterisation of 14/20 two-component systems (TCS) of GBS through the manipulation of the histidine kinase to prevent dephosphorylation of the cognate response regulator. This is an exciting approach that has been used to investigate the function of single TCS in other bacteria, but never at scale described here. The manuscript is extremely well written with novel findings related to a number of TCS and their networks.

I have a few minor comments

Throughout the manuscript the Streptococcus Walk homolog is called VikS, I cannot find this name anywhere in the literature. I would change to the VicK as has been published previously.

Line 126: In Supplementary table 3, what do the coloured boxes denote. Need a legend on the table. What is the CMI acronym?

Line 132: The Todd Hewitt Yeast Extract broth used is not standard (there are a number of different formulations). The composition is said in the M&M but would elaborate in the first use in the text eg. pH7.4 1% YE.

Line 133 Is the fold change Log2? Cannot directly compare to the graph in Fig 1C.

From the Fig 1C graph can see that the first 6 that autoregulate, but cannot see the 7th guessing VicK, but seems to be below the cut off specified.

Only 2/14 TCS could show increased phosphorylation on the RR? Find this result surprising and it is not commented on. Do you think that it is due to it being FLAG tagged disrupting function? Or that the Asp phosphorylation is too liable in 12/14? Please address.

Line 148 rich media, change to THY.

Line 150 delete highly

Have the P-values been analysed to give false discovery rate? Can the P-value give you the level of activation, the numbers for the P-value are so significant are they discriminatory?

Line 157: Why were mobile genetic elements excluded?

Line 190: What is IFCI?

Line 264: Please highlight that this is in *S. gordonii*.

Line 275: Could the improved growth of the SaeST133A be a consequence of the alleviation of the massive over expression of the proteins eg. Deletion of *bvaP* in the T133A also helps alleviate the growth deficiency?

Line 331: All methods are complementary, would tone down this. But the HK+ approach is exciting. But does require that point mutation can be introduced into the native HK – not put on plasmids.

Line 373: In *S. aureus*, the T389A Walk mutation is dominant over the repressive activity of YchH/Yycl, so not true for all HK with interacting regulators.

Line 418: Can the pG1 plasmid be described, or the sequence deposited. Went back through the literature and got to an unpublished plasmid.

Line 421: Please elaborate on the steps for mutant construction. How was the plasmid integrated (Broth vs plate), how long was the clone serially passaged at 30degC. Was the PCR screen for the mutant based to oligo binding at different temperatures?

Line 431 – What is being done in Geneious Prime – what settings were used? SNP calling threshold. What was the mapping program?

Line 513 change deposit.

REVIEWER COMMENTS

Reviewer #1 (Remarks to the Author):

The study by Claverie et al. aims to conduct a systematic examination of the two-component sensory network (TCS) in *Streptococcus agalactiae*. The authors undertake a transcriptomic analysis of mutants in the phosphatase motif of each histidine kinase (HK+), assuming that this mutation results in constitutive phosphorylation of the RR. This assumption is necessary due to the experimental conditions, where the signals activating most TCSs are unlikely to be present. To validate their hypothesis, the authors assess the effect of HK+ on TCS expression and observe positive feedback in 7 HK+. Additionally, they express 3xFlag-tagged RRs in strains with wild-type HK and HK+, observing the phosphorylated form of RR in only two of 14 RRs via Western blot. Transcriptomic analysis reveals transcriptional profile changes in 10 of 12 TCS examined, with variable regulon sizes and overlap between regulons. From this point, the manuscript focuses on analysing specific aspects of the regulation of three specific TCS: CovRS, the role of adhesin PbsP in the regulation of SaeRS and CovRS, and the activity of BreRS. These three stories are very specific and unrelated, and it is difficult to understand how they contribute to the overall understanding of the *S. agalactiae* TCS network.

We regret the misunderstanding. We did not assume that HK+ mutations would lead to the constitutive activation of signaling pathways. We have tested it.

This is the first time this overlooked approach has been tested systematically. To be honest, we were really amazed by the quality of the results. We did not expect such a resolution of signaling pathways for the majority of TCS. Still, there are limits, as not all TCS react similarly. We strongly believe that this study is of interest for researchers studying TCS signaling, not only in *S. agalactiae*. We hope that this approach will be widely adopted by the community.

Our study is data-rich, and we made choices.

Firstly, we performed a global analysis of HK+ mutants, highlighting both the generality of the approach and its limitations.

Secondly, we conducted targeted analyses of selected mutants, demonstrating the versatility of the HK+ approach. For these targeted analyses, we have chosen TCSs for which the HK+ approach provides unprecedented resolution of the regulons (SaeRS, BceRS) revealing unique biological insights related to the biology and virulence of *S. agalactiae*.

To summarize our contribution to the understanding of the *S. agalactiae* TCS network:

- Our study validates the conserved mechanism of HK phosphatase activity and provides the first comprehensive overview of the importance of this phosphatase activity *in vivo*.
- Our results provide the most detailed characterization to date for several TCS signaling pathways in *S. agalactiae*. Especially, activating TCS independently of environmental signals resolves the regulon with unprecedented precision for Sae, Bce, Vnc, Dlt, HK11030, and HK02290.

- Targeted analysis reveals an unexpected mechanism linking two major regulators of host-pathogen interactions (CovRS-SaeRS) through an adhesin (PbsP). This is the first time such TCS connection is observed with a virulence factor having an autocrine signaling function.
- Targeted analysis reveals a role of the BceRS system in cell envelope homeostasis. This challenges the current model of antibiotic resistance and, in addition, reveals a link with toxin expression, likely through toxin retention at the bacterial surface.

In this manuscript, we decided not to elaborate on the functions of highly specialized (VncRS, DltRS, HK11030, HK02290), global (VicRK, CovRS), or intermediately activated (CiaRH, RelRS) TCSs. The key points regarding these TCSs are:

- For specialized TCSs, the HK⁺ approach accurately determines the regulon, demonstrating for the first time their specialization. This is a significant advantage of the HK⁺ approach, as it circumvents the confounding effects of activating environmental signals, such as antimicrobial peptides in the case of DltRS
- For global TCSs, the HK⁺ approach is informative but not sufficient (as we show in the global analysis for the master regulator of virulence CovRS). These regulators require additional complementary methods due to their unique characteristics, such as essentiality and active repression.
- For intermediately activated TCSs, The HK⁺ approach successfully identifies the complex regulons of these systems, but the activation of the signaling pathway is limited. This limited activation, or absence thereof, is also observed in other systems such as Hss, Lia, and Pho, suggesting the presence of feedback mechanisms that warrant further investigation.

In summary, the systematic HK⁺ approach proves valuable for understanding differently responding types of TCSs and can be leveraged in future studies to decipher these additional signaling pathways mechanistically or functionally.

Major comments

Lines 110-116: In the case of WalkR, it is understood that compensatory mutations are generated when WalkR is mutated because its activity is essential, but in this case the mutation generated in VicST221A keeps the system constitutively active (phosphorylated) and the bacterium does not seem to like this and tries to compensate with other mutations. In general, all strains producing the mutant form of HK should be complemented with the wild-type form to exclude compensatory mutations in other genes.

We are confident in the genotype-phenotype relationships. All the mutants used in this study were sequenced (Supplementary Table S2). For the HK⁺ collection, 11 out of the 14 mutants have genomes identical to that of the parental wild-type (WT) strain, with the only difference being the introduced HK⁺ mutation. This excludes the possibility of compensatory mutations due to SNPs, deletions/insertions, or chromosomal rearrangements. For one additional strain (RelS), an extra thymidine is present in a stretch of six thymidines in a 3' UTR, which allows us to be confident in the genotype-phenotype relationship for RelS too.

We recognise that the secondary mutation in VicK_{T221A} is a limitation. This issue was specifically mentioned in the results section, with a reference addressing the common problem of secondary mutations in homologous WalRK systems (10.1038/ncomms1440) (lines 114-118). Nevertheless, the transcriptome of the VicK_{T221A} mutant clearly shows a conserved function compared to homologous systems in related bacteria (regulation of cell wall remodelling), indicating that the HK⁺ mutation indeed activates the signaling pathway.

As demonstrated in several species, the WalRK/VicRK systems are the only essential (or critical/conditionally essential) TCS. Both inactivation and constitutive activation impose a fitness cost, highlighting the tight regulation of cell wall metabolism during growth.

Importantly, the VicK_{T221A} mutant is unstable, accumulating additional compensatory mutations during growth (as shown in Supplementary Fig S1C). This complicates complementation, as it requires the preparation of competent cells and selection of transformants. Moreover, standard vector-based complementation cannot be used due to a gene dosage effect (multicopy vector), which is critical for regulatory proteins, and because VicK_{T221A} is a gain-of-function mutation. The alternative, repairing the chromosomal mutation, requires integrating a new construct at the locus, which inactivates the whole system, thereby creating conditions that select for new secondary mutations.

Overall, our results show the activation of a conserved signaling pathway by the VicK_{T221A} mutation in GBS. However, we did not pursue a detailed analysis of VicRK. The HK⁺ approach is informative but not sufficient for such global and nearly essential regulator. Its necessity for growth imposes constraints and requires complementary approaches, such as ChIP-sequencing, as done for the WalRK homologues in several species (10.1128/mbio.02262-23, 10.1038/ncomms14403, 10.7554/eLife.52088, 10.1073/pnas.0800247105).

Lines 134-136: To know if this induction is due to the vikS⁺ mutation and not to the off target mutation that the mutant has, the mutant would need to be complemented. RelRS pass the significance and the FC of 2? It seems that they are at the same point as vikRS in vikS⁺ in which they are not considered to be induced. Perhaps the significance of the TCS that pass it should be indicated in the graph with asterisks.

We thank the reviewer for pointing out this ambiguity. As suggested, we highlighted the seven auto-regulated TCS in Figure 1C. We have also added a reference to a new supplementary table 4E in the Fig. 1C legend. The supplementary table 4E now shows all the data (log₂ FC and p-adj) and can be used to view the TCS just below the thresholds (feedback of CiaRH, RgfAC in SaeS; BceRS in VicK) and the general perturbation associated with the activation of the two global regulators CovS and VicK (p-adj < 0.0001 for several TCS).

Line 168: Wouldn't there be 5? Hss, Rel, Sar and Bce but CovS also appears related to VikS. This interaction is shown in figure 2D but not mentioned in the text.

The reviewer is right. However, we have chosen not to mention CovS at this stage. We mentioned just afterwards in the results that we analysed the CovS mutant separately (now line 219). This is due to the particularities of the CovRS system, which is a repressor system. Please refer to the response to the next comment below concerning the dedicated section "Activation of the global repressor of virulence CovRS".

- Lines 206-208: The manuscript suddenly introduces a new mutant, CovRD53A, which cannot be phosphorylated. This sentence is not well understood. I think it means that inactivation of the RR activates the signalling pathways repressed by the active TCS. The way it is written, it seems very confusing to me. On the other hand, this comparison between the constitutively phosphorylated and non-phosphorylated form is very interesting and I think the manuscript should go in this direction rather than looking at specific aspects or phenotypes of CovRS and BceRS.

We acknowledge that we should have provided more information on CovRS in the initial manuscript, with its distinctive features. We have previously characterized the direct CovR regulon by RNA-sequencing and ChIP-sequencing, showing the complexity of this global repressor of virulence which is active in absence of environmental signal (10.1371/journal.pgen.1009761). In this manuscript, we included the previously described inactivated mutant (CovR_{D53A}) as well as our genome-wide binding data.

We have reworded paragraphs in the 'Activation of the global repressor of virulence CovRS' result section to better introduce CovR specificities and the CovR_{D53A} mutant (lines 262-274):

“On the other hand, a comparative analysis of repressed genes in CovS_{T282A} with the known CovR regulon shows the limitations of the HK⁺ approach for characterizing the CovRS system. Indeed, only 5 out of 14 repressed genes in the CovS_{T282A} mutant belong to the CovR regulon of 153 genes previously determined with loss of function mutants¹. This difference can be attributed to another specific features of the CovRS system, which is active in the absence of an environmental signal¹. The CovS_{T282A} transcriptome therefore suggests that CovR over-activation does not translate into increased repression of targeted genes.

To further analyse the CovRS system, we compared the transcriptome of the activated CovS_{T282A} mutant with that of the inactivated CovR_{D53A} mutant, which cannot be phosphorylated by CovS, and also included our genome-wide CovR binding analysis done by ChIP-seq¹. Side by side transcriptome comparison showed that inactivation of CovR activates the signaling pathway repressed by the active CovRS system, without a general inverse relationship in the CovS_{T282A} mutant (Fig. 3A), ...”

Regarding the suggestion to further characterize CovRS signaling, we are of course very interested in a detailed comparative analysis between phosphorylated and non-phosphorylated CovR. However, this would require a comparative analysis between different WT strains due to the plasticity and intra-species evolution of the CovR regulon (an additional characteristic of CovRS). We have generated and analysed a second CovS_{T282A} mutant in a different WT background. The analysis is complex, with strain-specificities and several types of promoters responding differently to CovR phosphorylation, as recently described for the homologous CovRS system of *Streptococcus pyogenes* using similar CovS_{T282A} mutation (10.1128/jb.00118-23, 10.1371/journal.ppat.1007354). These CovR mechanistic and evolutionary insights require a dedicated report.

Lines 213-214 The authors admit that the analysis with the non-phosphorylatable mutant is more informative than with the constitutively active mutant. This statement seems to contradict their whole approach.

This is specific to the CovRS system, as we initially mentioned: ‘The CovS_{T282A} mutation offers insight into the complexity of the CovR regulatory network but is less informative than the inactivation of the system.’.

Nevertheless, we have reworded the sentence to avoid any ambiguity (now lines 278-279): “Since the inactivation of the CovRS repressor system is more informative than its over-activation, we included the CovR_{D53A} transcriptome in the HK⁺ dataset.”

Line 220: The differences in haemolysis are not distinguishable in the photographs shown in the figure.

We have now provided high-resolution images for the revised version, rather than embedding them in a pdf. In addition, we modified the contrast of the images (homogeneously across all images) in Figure 3C to better highlight the hyper-haemolytic mutants (which lower the difference between the WT and the hypo-haemolytic mutants).

For your convenience, we have provided images of the complete plates (Granada and Columbia agar) below.

Line 225-226: This phenotype seems to be multifactorial and not only related to the regulation of Cyl genes, I do not understand how this result facilitates the understanding of the message of the manuscript.

Indeed, our results show that β -h/c toxin phenotypes are multifactorial. It was unexpected to identify several activated TCS with altered β -h/c toxin phenotypes. Moreover, not always correlating with *cyl* transcription. The HK+ approach not only identified activated TCS with altered toxin phenotypes, it allowed to characterize the basis of this altered phenotypes, revealing an original mechanism linking SaeRS with CovRS via the PbsP adhesin (with β -h/c phenotypes dependent of *cyl* transcription) and the function of the BceRS system in cell envelope homeostasis (with β -h/c phenotypes independent of *cyl* transcription). Altogether, we demonstrate the versatility of the HK+ approach to identify regulatory networks and to characterize individual systems, revealing unique biology directly linked to the virulence of *S. agalactiae*.

Minor comments

Line 118, reproducible, change for reproducible

Done

Line 119: “and two having specific phenotypes” They would actually have a fitness disadvantage.

We have modified the sentence accordingly (now line 121: “... the *VicK*_{T221A} and *SaeS*_{T133A} have fitness defect.”).

Lines 201-202: If the genes repressed by VicS are 17 and by Cov are 14, the total number of repressed genes is 31 and not 32.

They are genes repressed by more than one system, so the total number is not an addition. The 32 genes are listed in Supplementary Table S4G and are also illustrated in Fig. 2B. In addition to VicK and CovS, one gene is significantly repressed in VncS, 2 by RelS, 2 by CiaH, 1 by RS06940, and 1 by BceS.

We have amended the sentence (now line 258) for: ‘Almost all repressed genes are regulated by the two global regulators VicK (17 genes) and CovS (14 genes).’

Line 179: There is no section E in figure 2.

Thank you. The spelling mistake has been changed (Fig 2D).

Line 220: The differences in haemolysis are not distinguishable in the photographs shown in the figure.

Please refer to the response to the identical comment above.

Figure 1C: HK⁺ mutants should be reported as SaeS⁺, etc. A consistent criterion in nomenclature should be maintained in all tables, figures and text. This would facilitate reading. To be precise and consistent, we have homogenized the names by indicating the mutation for each variant (e.g. SaeS_{T133A}) in the text, figures and tables. When we refer to the approach or to the collection of mutants, we call it HK⁺.

- Line 542: "...time (F = Dx / DWT mean)" switch for "(Fitness = Doubling time x / Doubling time WT mean)".

As pointed by reviewer 2, we corrected the formula for "(Fitness = Doubling time WT mean / Doubling time X)".

- Line 543: "(dashes)" switch for "(dashes lines)".

Done.

Figure 2A: Indicate that 1000x, 100x, 10x refers to the FC.

Done.

The genes represented in the CovR node are induced in a CovR mutant, so they are repressed by CovR, while the rest of the nodes represent genes induced by the active form of HK (HK⁺). The genes of the CovR node should be shown in a different colour to distinguish repressed genes from induced genes.

Thank you for the suggestion. We have highlighted CovR-repressed genes using a different colour.

Figure 4F: The figure should reflect what is mentioned in the text that the adhesin may be interacting with CovS or coregulatory proteins or inducing surface perturbations specifically detected by the CovRS system.

Thank you for the suggestion. We added the Stk1/CovS/Abx1 regulatory proteins on Fig 4F.

Supplementary Figure S2: HK11030 and VncS results are in Figure 2, remove them to avoid duplication.

The Supplementary Figure 2 provides a comprehensive view of the mutants. We believe it is important to have all transcriptome side by side for direct comparison. We would have liked to include the entire ensemble as a main figure, but space constraints prevented us from doing so. Therefore, we illustrated the Fig. 2 with HK11030 and VncS, and used the volcano in Fig. 3A for CovS and in Fig. 4A for SaeS.

Reviewer #2 (Remarks to the Author):

In this manuscript, the authors made phosphatase KO mutations in 14 sensor kinases of two-component systems in Group B strep and characterized their effect on growth, autoregulation, and cross-regulation. The key results are:

1. HK11050-RR11055 system and RelRS were cross-regulated by VikRS and CiaRH TCS, respectively.
2. The SaeRS TCS activates the CovRS TCS through the surface protein PbsP.
3. BceRS activation confers resistance to nisin and gramicidin but makes the cells more sensitive to colistin. Of the BceRS regulon, BceD and BceO were not required for the nisin resistance, but BceP was required to some extent.

Based on these results, the authors concluded that the phosphatase KO mutation approach can be broadly applied to studying TCS in any species.

The methods employed are robust, and most data are of high quality and clear. In general, the authors' conclusions are well-supported by the presented data. The utility of the phosphatase KO approach was demonstrated for 11 TCSs. However, I still have some concerns:

1. It remains unclear whether the phosphatase KO mutation approach can be universally applied. Only two RRs out of 14 showed elevated phosphorylation in the phos-tag assay (Fig. 1D), and the HK⁺ mutation did not significantly affect the activity of LiaS and PhoS (Fig. S2). Our systematic analysis shows that the approach can be generally, but not universally, applied. Our results validate the conserved mechanism of phosphatase activity for almost all HisKA/KA3, but show variable activation of the signaling pathway depending on each TCS. Systematic approaches in other species are now required to define “rules” to predict how an uncharacterized system will respond to the selective inactivation of the HK phosphatase activity.

We believe that reporting the negative results for LiaS and PhoS is important. The lack of published negative results is one of the main reasons for our “naïve” systematic approach. With only a few individual examples published where the approach worked, it was impossible to assess its general applicability. It remains now to be determined why LiaS and PhoS (and to a lesser extent, HssS) are not responding. In the discussion, we suggested that negative feedback loops by small proteins of the regulon could lock LiaS and PhoS in a kinase-deficient conformation, thereby negating the effect of HK⁺ mutations (lines 479-482). Alternatives should also be explored, especially the *in vivo* significance of phosphatase activity and the high rate of spontaneous dephosphorylation of the cognate regulators compared to other RRs.

As also noted by Reviewer 3, we were surprised by the low number of regulators showing increased phosphorylation by Phos-Tag. However, due to the experimental conditions (ectopic expression of an epitope-tagged variant in an HK⁺ mutant containing the WT regulator), we couldn't draw conclusions for most regulators (in contrast to the transcriptomic approach). The Phos-tag experiment has several limits, such as competition between the WT and ectopically expressed variants (the latter of which is not guaranteed to be functional), variability in phosphorylated aspartate stability, and differences in expression levels and stability of the expressed epitope-tagged regulator despite being tested under the same conditions. Consequently, the phosphorylation of each RR must be assessed individually using specific antibodies against the native form and considering their specific rates of spontaneous dephosphorylation.

The initial discussion included one sentence (now lines 487-491) on the need to specifically test each system individually: “However, detailed analysis is required to decipher phosphorylation dynamics in each phosphatase-deficient HK⁺ and correlate *in vivo* RR phosphorylation with regulatory network activation, considering variable factors like the source of RR phosphorylation (kinase activity of the HK⁺ variant, cross-talk by other HK, small metabolites) and specific spontaneous dephosphorylation rates of the labile aspartate residues”.

We have now added a sentence in the results section (lines 158-160) summarizing the limitations of systematic quantification by Phos-Tag: “However, due to competition between WT and epitope-tagged regulators and variability in the stability of phosphorylated aspartate, no conclusions could be drawn for most regulators. This highlights the need to quantify the level of phosphorylation using specific antibodies directed against each native RR”.

2. The lack of a significant relationship between the expressions of direct targets of CovRS in CovR (inactive) and CovS (active) mutants is puzzling (Fig 3B). One would expect an inverse relationship, but this was not observed.

The reviewer is right. However, we were not surprised, considering the complexity of the CovRS regulatory network and the published analyses on the homologous CovRS system with

similar inactive and active mutants in *Streptococcus pyogenes* (10.1128/jb.00118-23, 10.1371/journal.ppat.1007354).

Following the remark of reviewer 1, we have reworded paragraphs in the ‘Activation of the global repressor of virulence CovRS’ result section (lines 262-274) to better introduce CovR specificities and the CovR_{D53A} mutant:

“On the other hand, a comparative analysis of repressed genes in CovS_{T282A} with the known CovR regulon shows the limitations of the HK⁺ approach for characterizing the CovRS system. Indeed, only 5 out of 14 repressed genes in the CovS_{T282A} mutant belong to the CovR regulon of 153 genes previously determined with loss of function mutants¹. This difference can be attributed to another specific features of the CovRS system, which is active in the absence of an environmental signal¹. The CovS_{T282A} transcriptome therefore suggests that CovR over-activation does not translate into increased repression of targeted genes.

To further analyse the CovRS system, we compared the transcriptome of the activated CovS_{T282A} mutant with that of the inactivated CovR_{D53A} mutant, which cannot be phosphorylated by CovS, and also included our genome-wide CovR binding analysis done by ChIP-seq¹. Side by side transcriptome comparison showed that inactivation of CovR activates the signaling pathway repressed by the active CovRS system, without a general inverse relationship in the CovS_{T282A} mutant (Fig. 3A), ...”

We are very interested in a detailed comparative analysis between phosphorylated and non-phosphorylated CovR. However, this would require a comparative analysis between different WT strains due to the plasticity and intra-species evolution of the CovR regulon (an additional characteristic of CovRS). We have generated and analysed a second CovS_{T282A} mutant in a different WT background. The analysis is complex, with strain-specificities and several types of promoters responding differently to CovR phosphorylation, as described in *Streptococcus pyogenes* (10.1128/jb.00118-23, 10.1371/journal.ppat.1007354). These CovR mechanistic and evolutionary insights require a dedicated report.

3. In Fig. 4D, it is necessary to confirm the proper expression of the C-terminal deletion mutant of PbsP (PbsPdC-ter).

We confirmed the expression of the PbsP Δ Cter variant using FACS analysis. We used an anti-PbsP serum produced in mouse on paraformaldehyde fixed bacteria. We analyzed the fluorescence at the cell surface with secondary phycoerythrin(PE-A)-conjugated goat anti-mouse IgG.

The figure below summarizes the results obtained with two biological replicates, including negative controls (CTR-) carried out for each strain, following the same procedure but without incubation with anti-PbsP serum.

These results first validate the overexpression of PbsP in the SaeS_{T133A} mutant compared to the WT strain and the absence of PbsP in the SaeS_{T133A} Δ *pbsp* double mutant.

Deletion of the C-terminal part of PbsP, which includes the deletion of the LPxTG cell-wall anchoring motif, is associated with retention of a significant fraction of the overexpressed PbsP Δ Cter variant at the surface of the SaeS_{T133A} mutant. Compared to the WT strain, the relative fluorescence signal is 9.70 time superior in the SaeS_{T133A} mutant and 3.32 time superior in the SaeS_{T133A} PbsP Δ Cter mutant.

Deletion of the internal part of the PbsP adhesin leaves only the signal peptide and the C terminal cell wall anchoring motif. As a result, the SaeS_{T133A} PbsP Δ intern mutant lost almost all reactivity with anti-PbsP serum.

We hope that these results address the legitimate concern of the reviewer. However, we would prefer not to incorporate these results in the manuscript. We aim to strengthen them by

identifying the minimal portion of PbsP necessary to induce CovR signaling and by characterizing the interaction of PbsP with the CovS/Abx1/Stk1-CovR signaling complex. All this requires considerable work and a separate manuscript.

4. The section from lines 317 to 328 is difficult to follow and understand. I do not see evidence supporting their claims such as “the BceRS system is active in the absence of drugs,” “The BceRS response is adaptative ..”, “... the response is constrained by its cost against antimicrobial peptides”, or “ the BceRS system actively monitors and adjust surface-exposed lipid II metabolites.” It would be more suitable for the Discussion section rather than the Results.

We recognize that placing all intermediate conclusions in a single paragraph at the end complicates the reading. We have now moved the intermediate conclusions to the corresponding figures within the presentation of the results (now lines 362-417).

Fig 5C: “These results show that BceRS activation confers protection against structurally unrelated drugs targeting lipid II intermediates at the cost of increased susceptibility to antimicrobial peptides.”

Fig 5D: “ These results show that the BceRS system is constitutively active in the absence of drugs and that the basal activity is necessary and sufficient to counteract the effects of sub-inhibitory concentrations of nisin.”

Fig. 5E: “Altogether, these results show that individual genes do not provide drug resistance and suggests that the BceP extracellular S41 protease has a buffering role when the pathway is activated ...”

Fig. 5F: “These results show that the BceRS response is adaptative and suggests that the BceRS system actively monitors and adjust surface-exposed lipid II metabolites, rather than directly detoxifying drugs or drug-lipid II complexes.”

The last point is included in the discussion with reference to the literature. We now also moved in the discussion (now lines 510-513): “It is also interesting to note that the activation of the BceRS-BceAB system impacts the retention or secretion of the β -h/c toxin, potentially as a result of the interaction between the polyene backbone of the toxin and the membranes²⁻⁴, suggesting functional links between cell envelope homeostasis, drug resistance, and virulence.”

Other minor comments

1. Please verify the correctness of the fitness formula (D_x/D_{wt}) in Fig 1B.

Thank you for pointing out the error. We corrected for “(Fitness = Doubling time WT mean / Doubling time X)”

2. Line 119 “ .. two having specific phenotypes..”: It would be helpful if the authors specify the phenotypes.

We have modified the sentence accordingly to avoid any ambiguity (“... the VicK_{T221A} and SaeS_{T133A} have fitness defect.”).

3. Line 178-179, 181: There is no Fig 2E. May be 2D?

Right. We have corrected for 2D.

4. Line 250: It is unclear what the authors mean by “dynamic” of the response.

We mean the signal-dependent kinetics of the response. The increased expression of the TCS should impact the response, either by sustaining a longer reaction to a transient activating signal, by promoting a new equilibrium that allows for adaptation to variations in signal concentration, or by enhancing the overall sensitivity to changes in the signal. In the text (now lines 334), we have replaced “dynamic response” with “for controlling the kinetics of the signal-dependent response”.

5. Line 411: Hepes pH7.4 -> HEPES pH 7.4

Done.

6. Line 487: 18h -> 18 h

Done.

7. Line 504: I am not sure how Supplementary Table S4 is related to the cloning step. The table number has been corrected in the text (S5 = oligonucleotides, S6 = cloning).

Reviewer #3 (Remarks to the Author):

The manuscript of Claverie et al. describes the characterisation of 14/20 two-component systems (TCS) of GBS through the manipulation of the histidine kinase to prevent dephosphorylation of the cognate response regulator. This is an exciting approach that has been used to investigate the function of single TCS in other bacteria, but never at scale described here. The manuscript is extremely well written with novel findings related to a number of TCS and their networks.

I have a few minor comments

Throughout the manuscript the Streptococcus WalK homolog is called VikS, I cannot find this name anywhere in the literature. I would change to the VicK as has been published previously. Right. Thank you. We have changed the name to VicK, and consequently to VicR, in the text, figures and tables.

Line 126: In Supplementary table 3, what do the coloured boxes denote. Need a legend on the table. What is the CMI acronym?

Colours indicate increased susceptibility compared with the WT strain. A legend has been added.

Thank you for pointing out the typo: CMI has been replaced by MIC (minimum inhibitory concentration).

Line 132: The Todd Hewitt Yeast Extract broth used is not standard (there are a number of different formulations). The composition is said in the M&M but would elaborate in the first use in the text eg. pH7.4 1% YE.

Done.

Line 133 Is the fold change Log₂? Cannot directly compare to the graph in Fig 1C.

These were fold change (FC > 2). We now clarify in the text (now lines 136) : “... indicated by a significant fold change greater than two relative to the WT strain (log₂ FC > 1; p-adj < 10⁻⁴) for the HK and RR gene, ...”

From the Fig 1C graph can see that the first 6 that autoregulate, but cannot see the 7th guessing VicK, but seems to be below the cut off specified.

Indeed, VicK is the 7th TCS, just above the thresholds.

As suggested by reviewer 1, we highlighted the seven auto-regulated TCS in Figure 1C. We have also added a reference to a new supplementary table 4E in the Fig. 1C legend. The supplementary table 4E now shows all the data (log2 FC and p-adj) and can be used to view the TCS just below the thresholds (feedback of CiaRH, RgfAC in SaeS; BceRS in VicK) and the general perturbation associated with the activation of the two global regulators CovS and VicK (p-adj < 0.0001 for several TCS).

Only 2/14 TCS could show increased phosphorylation on the RR? Find this result surprising and it is not commented on. Do you think that it is due to it being FLAG tagged disrupting function? Or that the Asp phosphorylation is too liable in 12/14? Please address.

Thank you for the remark. Indeed, we were surprised by the low number of regulators showing increased phosphorylation by Phos-Tag (also pointed out by reviewer 2). We initially discussed the technical hurdles in the original draft but deleted them due to space considerations. We have now added a sentence in the results section summarizing the limitations of systematic quantification by Phos-Tag (now lines 157-160): “However, due to competition between WT and epitope-tagged regulators and variability in the stability of phosphorylated aspartate, no conclusions could be drawn for most regulators. This highlights the need to quantify the level of phosphorylation using specific antibodies directed against each native RR.”

The initial discussion also included one sentence on the need to test each system individually and specifically (now lines 487-491): “However, detailed analysis is required to decipher phosphorylation dynamics in each phosphatase-deficient HK⁺ and correlate *in vivo* RR phosphorylation with regulatory network activation, considering variable factors like the source of RR phosphorylation (kinase activity of the HK⁺ variant, cross-talk by other HK, small metabolites) and specific spontaneous dephosphorylation rates of the labile aspartate residues”.

Line 148 rich media, change to THY.

Done.

Line 150 delete highly

Done.

Have the P-values been analysed to give false discovery rate? Can the P-value give you the level of activation, the numbers for the P-value are so significant are they discriminatory?

P-values were indeed corrected for multiple testing, using Benjamini-Hochberg adjustment method, that controls the FDR.

We apologize for the inaccurate wording of the section. The p-value only gives the conclusion of the test and does not indicate the activation levels (provided by the FC, highlight in the following panel 2B). The aim of the initial analysis (Fig. 2A and supplementary Fig. 2) was precisely to highlight the exceptional p-values, without considering the activation level. The transcriptomic responses are so significant for 6 TCS that they enable the resolution of the regulon with unprecedented detail and precision.

We have rewritten this section to clarify (now lines 166-173):

“Statistical analysis of the differentially expressed genes (DEGs: Supplementary Table S4D) grouped the HK⁺ mutants into three main categories based on the adjusted p-values. Six HK⁺ mutants (HK11030_{T245A}, VncS_{T245A}, SaeS_{T133A}, BceS_{V124A}, HK02290_{H188A}, and DltS_{T184A}) show DEGs associated with striking statistical significance (p-adj < 10⁻²⁵⁰), revealing the activated regulons with high resolution (Fig. 2A and Supplementary Fig. S2). Four additional mutants (RelS_{T208A}, CiaH_{T228A}, VicK_{T221A}, and CovS_{T282A}) show DEGs with lower statistical significance (p-adj > 10⁻¹⁵⁰), suggesting complex regulons or intermediate TCS activation (Supplementary Fig. S2).…”

Line 157: Why were mobile genetic elements excluded?

We are now systematically analysing mobile genetic elements (MGEs) on a separate basis.

Some MGEs are transcribed at low basal level, have specific mechanism of regulation, and are unstable. For example, we observed instances of MGE loss in only one of the biological replicates in previous transcriptomic analyses, which led to the incorrect identification of significant differential expression through global analysis. In addition, we also observed significant variability for MGE-encoding genes only in the WT strains between independent experiments (batch effects). We also previously reported that some MGEs appears silenced by the CovRS system, which necessitated separate analyses (10.1371/journal.pgen.1009761).

Line 190: What is IFCI?

It should have been the absolute value symbol. We corrected the typo for |FC|.

Line 264: Please highlight that this is in *S. gordonii*.

Done.

Line 275: Could the improved growth of the SaeST133A be a consequence of the alleviation of the massive over expression of the proteins eg. Deletion of *bvaP* in the T133A also helps alleviate the growth deficiency?

Yes, indeed. The growth defect of the SaeS T133A mutant is partially alleviated by the deletion of *pbsP* or *bvaP*. The two genes appear to contribute equally to the growth defect of SaeS T133A. We have not specifically mentioned the growth of the *bvaP* mutant in the main text, but growth curves were included in Supplementary Fig. 4A. Nevertheless, we changed the Supp Fig. S4 title from “The fitness defect in SaeS T133A is caused by the adhesin PbsP” to “PbsP and BvaP contribute to SaeS T133A fitness defect.”

Line 331: All methods are complementary, would tone down this. But the HK+ approach is exciting. But does require that point mutation can be introduced into the native HK – not put on plasmids.

Right, this is a (powerful) complementary approach. We tone down by changing (now line 420) “a method of choice” by “Our systematic analysis highlights the benefits of the HK⁺ gain-of-function approach to characterize TCS signaling”.

Line 373: In *S. aureus*, the T389A WalK mutation is dominant over the repressive activity of YycH/YycI, so not true for all HK with interacting regulators.

Right. Thank you for pointing that out. We have added a corresponding sentence to the discussion (lines 482-484): “However, the presence of auto-inhibitory proteins is not a sufficient condition for preventing activation, as demonstrated by VicK, which is inhibited by YycH/YycI but still activated by an HK⁺ mutation.”

Line 418: Can the pG1 plasmid be described, or the sequence deposited. Went back through the literature and got to an unpublished plasmid.

The pG1 plasmid is, or is almost identical to, the pG+host5 vector described in 1993 by Biswas et al. (Biswas, I., Gruss, A., Ehrlich, S. D. & Maguin, E. High-efficiency gene inactivation and replacement system for gram-positive bacteria. *J Bacteriol* **175**, 3628-3635). However, we cannot precisely track down the origin of the plasmid. It appears that the two plasmids were constructed independently in pG+host4 vector, one using the pBR322 origin of replication (pG+host5) and the other the pUC18 origin of replication (pG1). Both origins being the pMB1 *rep* for replication at 37°C in *E. coli*. The other main characteristics are a thermosensitive origin of replication derived from pWV01, for rolling circle replication in gram-positive and gram-negative, and an erythromycin resistance marker.

We added in the manuscript the reference for the pG+host5 vector with the sentence (line 534): “pG1, a thermosensitive shuttle vector similar to the pG+host5 vector.”

We also added the reference in the supplementary Table S6, listing all vector with their main characteristics, altogether with the reference for the pTCV-lac transcriptional reporter vector and for the pTCV-tetO inducible expression vector.

We are delighted to share sequences and plasmids upon request (with standard academic MTA). We are attempting to ascertain the vector's origins before deposition (Addgene).

Line 421: Please elaborate on the steps for mutant construction. How was the plasmid integrated (Broth vs plate), how long was the clone serially passaged at 30degC. Was the PCR screen for the mutant based to oligo binding at different temperatures?

For PCR screening, we used primers with the mutations at their 3' extremities (Supp Table S5). We generally introduce 3 SNPs to make the targeted codon change (with neutral SNP in the penultimate codon if necessary), which is sufficient to ensure specificity at our standard annealing temperature (58°C).

As requested, we detailed mutant construction in the Material and Method section (lines 538-566):

“ Mutant construction in GBS was performed through a three-step process involving episomal replication, chromosomal integration, and vector loss. Initially, pG vectors were introduced into GBS via electroporation, and transformants were selected on THY agar supplemented with 5 µg/ml erythromycin at 30°C, the permissive temperature for episomal replication. After 24-36 hours of growth, two single transformant colonies were isolated on THY agar containing erythromycin and incubated another 24-36 hours at 37°C, the non-permissive temperature for vector replication, thereby promoting chromosomal integration of the vector at the targeted locus through homologous recombination. Subsequently, rare colonies that had integrated the vector into the chromosome were isolated on THY agar with erythromycin at 37°C for 16 hours. The isolated integrants were inoculated into 10 ml of THY medium without antibiotic, incubated at 30°C to activate the rolling circle origin of replication, and subjected to serial subculture twice daily. By day 3, cultures were diluted (typically 10^{-5}), spread onto THY and Columbia agar supplemented with 10% horse blood (BioMerieux), and incubated at 37°C. Isolated colonies (n = 24-48) were picked into 150 µl of THY in 96-well plates, incubated for 4-6 hours at 37°C, and replica-plated using a 96-pin replicator (Boeckel Scientific) onto THY agar plates with and without erythromycin. After 16 hours at 37°C, erythromycin-susceptible colonies that had lost the vector were identified and confirmed by discriminatory PCR (MyTaq HS - Biorline) using specific oligonucleotides with the expected mutation at their 3' extremity (Supplementary Table S5) to distinguish mutant from wild-type genotypes. Mutants were then isolated on THY at 37°C, single colonies inoculated in THY incubated at 37°C for 16 hours, centrifuged and resuspend in 20% glycerol for long-term storage at -80°C.”

Line 431 – What is being done in Geneious Prime – what settings were used? SNP calling threshold. What was the mapping program?

Geneious Prime uses a private mapper, equivalent to BMap and Bowtie2, that we used with default parameters (Up to 5 iteration, Minimum mapping quality 30 (Phred score), Medium/low sensitivity) using the BM110 reference genome. SNP calling was done with the Geneious private software, with minimum variant frequency 0.25, Maximum Variant P-value 10^{-6} , Minimum strand-Bias P-value 10^{-5} .

We have added the following to the Materials and Methods section (lines 572-574):

“... using default parameters (mapping: up to 5 iteration, minimum mapping quality Phred score 30, medium/low sensitivity; SNP calling: minimum variant frequency 0.25, maximum variant P-value 10^{-6} , minimum strand-bias P-value 10^{-5} , with visual inspection for coverage and validation of SNPs).”

Line 513 change deposit.

Done. Change for “loaded”

Reviewer #1 (Remarks to the Author):

I understand and accept the authors' argument that the initial analysis generates a huge amount of data and that they are looking at very specific aspects of ECT regulation to continue the study. I agree that the regulation and results related to BceRS are very interesting. The coordination of regulation between SaeRS and CovRS is also very interesting. Still, what they analyse with BceRS is not at all related to what is analysed between Sae/Cov, and in my opinion this lack of relationship makes the article confusing to read and makes it difficult to draw general conclusions.

An example that well illustrates the lack of clarity in summarising the article's main conclusions is that in the authors' response to my first comment, they highlight the four main contributions of the article. Of these, the first two contributions do not appear explicitly in the article's abstract, and I think they should.

- Our study validates the conserved mechanism of HK phosphatase activity and provides the first comprehensive overview of the importance of this phosphatase activity in vivo.
- Our results provide the most detailed characterization to date for several TCS signaling pathways in *S. agalactiae*. Especially, activating TCS independently of environmental signals resolves the regulon with unprecedented precision for Sae, Bce, Vnc, Dlt, HK11030, and HK02290.

That said, I think the article has very interesting aspects and that the authors have taken seriously the response to the suggestions made by the reviewers, and the article has improved significantly in this new version.

Minor point

There is an error in the description of panel A in figure 2. A. HK11030T245A is the left panel and VcnST245A is the right panel.

Reviewer #2 (Remarks to the Author):

This is a revised manuscript studying GBS TCS systems with phosphatase-deficient histidine kinases. The authors have properly addressed my previous concerns through their revisions, and I have no further comments except for two minor ones listed below:

Line 647: Change 'promoter' to 'terminator.'

Fig. 5D–F: Although evident, please label the Y-axis with 'OD600' and the X-axis with 'Time (hours).'"

Reviewer #3 (Remarks to the Author):

I am happy with the changes to the updated manuscript and the authors have directly addressed my points raised. One comment is that the mapping of Illumina reads mapped to a reference would not completely identify all potential genomic changes (Reviewer #1 Major

comments Lines 110-116 - rebuttal). To get past the need for complementation, I would suggest complete genome assembly. This is becoming far more affordable - and error free genome assemblies from just nanopore reads have been shown (<https://www.ncbi.nlm.nih.gov/pmc/articles/PMC11170131/>).

REVIEWERS' COMMENTS

Reviewer #1 (Remarks to the Author):

I understand and accept the authors' argument that the initial analysis generates a huge amount of data and that they are looking at very specific aspects of ECT regulation to continue the study. I agree that the regulation and results related to BceRS are very interesting. The coordination of regulation between SaeRS and CovRS is also very interesting. Still, what they analyse with BceRS is not at all related to what is analysed between Sae/Cov, and in my opinion this lack of relationship makes the article confusing to read and makes it difficult to draw general conclusions.

An example that well illustrates the lack of clarity in summarising the article's main conclusions is that in the authors' response to my first comment, they highlight the four main contributions of the article. Of these, the first two contributions do not appear explicitly in the article's abstract, and I think they should.

- Our study validates the conserved mechanism of HK phosphatase activity and provides the first comprehensive overview of the importance of this phosphatase activity in vivo.
- Our results provide the most detailed characterization to date for several TCS signaling pathways in *S. agalactiae*. Especially, activating TCS independently of environmental signals resolves the regulon with unprecedented precision for Sae, Bce, Vnc, Dlt, HK11030, and HK02290.

We thank the reviewer for his/her positive feedback.

We have now edited the abstract to highlight the two points mentioned above.

That said, I think the article has very interesting aspects and that the authors have taken seriously the response to the suggestions made by the reviewers, and the article has improved significantly in this new version.

Minor point

There is an error in the description of panel A in figure 2. A. HK11030T245A is the left panel and VcnST245A is the right panel.

Thank you. The typo has been corrected in the legend.

Reviewer #2 (Remarks to the Author):

This is a revised manuscript studying GBS TCS systems with phosphatase-deficient histidine kinases. The authors have properly addressed my previous concerns through their revisions, and I have no further comments except for two minor ones listed below:

Line 647: Change 'promoter' to 'terminator.'

Done. Thank you for noticing the error.

Fig. 5D–F: Although evident, please label the Y-axis with 'OD600' and the X-axis with 'Time (hours).'"

Done.

Reviewer #3 (Remarks to the Author):

I am happy with the changes to the updated manuscript and the authors have directly addressed my points raised. One comment is that the mapping of Illumina reads mapped to a reference would not completely identify all potential genomic changes (Reviewer #1 Major comments Lines 110-116 - rebuttal). To get past the need for complementation, I would suggest complete genome assembly. This is becoming far more affordable - and error free genome assemblies from just nanopore reads have been shown (<https://www.ncbi.nlm.nih.gov/pmc/articles/PMC11170131/>).

The reviewer is right to say that short reads are limited in detecting genomic duplication and structural chromosome rearrangement. In addition to SNP analysis, we systematically check for the absence of duplicated chromosomal regions (by visual inspection of coverage along the chromosome) and the presence of breaks in coverage indicating a large deletion, insertion or rearrangement (by visual inspection of alignments with SNPs at a frequency of around 25-75%). We don't yet have any experience of long-read sequencing, whose first generations were very limited for low GC% genomes (the case of *S. agalactiae*). The most recent generations mentioned seem promising and we are interested in evaluating them on our low GC% genomes.